

# A data-model synthesis to explain variability in calcification observed during a CO$_2$ perturbation mesocosm experiment

Shubham Krishna[1] and Markus Schartau[1]

[1]GEOMAR Helmholtz Centre for Ocean Research Kiel, Düsternbrooker Weg 20, 24105 Kiel, Germany

*Correspondence to:* Shubham Krishna (skrishna@geomar.de) and Markus Schartau (mschartau@geomar.de)

**Abstract.** A series of studies were conducted during the last two decades to investigate effects of ocean acidification (OA) on phytoplankton physiology, plankton ecology, and biogeochemical dynamics of marine ecosystems. Among those studies are experiments with tanks or bags called mesocosms, with some enclosed water volume that typically comprised a natural plankton community found in the surrounding environment. The Pelagic Ecosystem CO$_2$ - Enrichment Study PeECE-I experiment is one such study, where mesocosms were perturbed and exposed to different carbon dioxide (CO$_2$) concentrations to determine responses in growth dynamics of the coccolithophorid *Emiliania huxleyi*, a marine calcifying algae. The data from replicate mesocosms of PeECE-I show some natural variability and significant differences were revealed in the accumulation of particulate inorganic carbon (PIC) between mesocosms of similar CO$_2$ treatments.

In our study we reanalyse PeECE-I data and apply an optimality-based model approach to understand most of the variability observed, with major focus on total alkalinity (TA) and calcification. We explore how much of the observed variability in data can be explained by variations of initial conditions and by the effect of CO$_2$ perturbations. According to our model approach, changes in cellular calcite formation are resolved at the organism-level in response to variations in CO$_2$. With a data assimilation (DA) method we obtain three distinctive ensembles of model solutions, with low, medium and high calcification rates. Optimised values of initial conditions turned out to be correlated with estimates physiological model parameters. The spread of ensemble model solutions captures most of the observed variability, corresponding to the combinations of parmeter estimates. Optimised model solutions of the high CO$_2$ treatment are shown to systematically overestimate observed PIC production. Thus, the simulated CO$_2$ effect on calcification is likely too weak. At the same time our model results yield large differences in optimal mass flux estimates of carbon and of nitrogen even between mesocosms exposed to similar CO$_2$ conditions.

## 1  Introduction

Much knowledge about growth and mortality of phytoplankton has been inferred from experiments where environmental factors like light, temperature, and nutrient availability have been predominantly controlled, e.g. in laboratory experiments with batch cultures or with chemostats. Typically, these experiments are designed to determine a physiological response to variations of a single factor, e.g. explaining changes in photosynthetic rate when exposed to different light conditions (e.g. Platt et al., 1977; Marra and Heinemann, 1982; Lewislg and Smith, 1983; Geider et al., 1985; Harrison and Platt, 1986; Harding Jr et al., 1987). Many laboratory experiments are performed with monocultures, with the advantage that physiological





responses may then become well expressed in measurements while variability between replicates or even between repeated experiments should remain low. In this context a series of laboratory studies with monocultures of calcifying coccolithophores were conducted to investigate responses in calcification to variations in carbonate chemistry, often with *Emiliania huxleyi*, (e.g. Zondervan et al., 2002; Iglesias-Rodriguez et al., 2008; Langer et al., 2009; Barcelos e Ramos et al., 2010). These studies

were motivated by the expectation that the observed trend in ocean acidification (OA) will affect calcifying algae and that their physiology is likely sensitive to the seawater's calcite saturation state (Feely et al., 2004; Orr et al., 2005).

The repeated laboratory OA experiments showed ambiguous responses in calcification to variations in carbon dioxide ($CO_2$) concentrations and Findlay et al. (2011) pointed out that differences in laboratory methodology, but also details in experimental design, are likely the reason for the large observed variability in *E. huxleyi* responses to changes in carbonate chemistry.

Similarly, Engel et al. (2014) stressed that variations in the observed ratio between particulate inorganic carbon and particulate organic carbon (PIC:POC ratio) increase with the decrease of measured relative growth rates, depending on whether "low" growth conditions were balanced (as achieved with chemostats) or resulted from unresolved transient nutrient-limitation effects in batch cultures. This ongoing discussion is accompanied by the question of how representative the outcomes of monoculture laboratory experiments are, to allow for reliable future projections of OA effects on oceanic calcification rates of

coccolithophores and on possible climate feedbacks.

If we seek to make inference about future changes in calcification under oceanic conditions, experimental data are needed that consider more realistic environmental conditions with a natural phytoplankton community that may include calcifying algae like *E. huxleyi*. This was approached with a series of mesocosm experiments, where enclosed seawater volumes were exposed to different $CO_2$ concentrations, e.g. Pelagic Ecosystem $CO_2$ Enrichment (PeECE) studies (Riebesell et al., 2008).

In contrast to monoculture laboratory experiments, $CO_2$ perturbation mesocosm experiments yield "net" community response signals that are anticipated to be more indicative for possible future changes in oceanic calcification of coccolithophores. Replicate mesocosms with similar initial nutrient, as well as initial dissolved inorganic carbon (DIC) concentrations typically show comparable temporal response patterns, i.e. an exponential growth phase until nutrients become depleted and a post-bloom period where chlorophyll *a* concentrations decline. However, replicate mesocosms that all included *E. huxleyi* exhibited

large deviations in calcification responses, thereby altering carbonate chemistry. Such variability was well reflected in total alkalinity (TA) measurements of the PeECE-I experiment (Delille et al., 2005). Furthermore, during PeECE-I it happened that high and low calcification rates were revealed within each of the three different $CO_2$ treatments. To find enhanced variability in calcification in mesocosm experiments is comprehensable and can be attributed to the likely mixture of superimposed responses of multiple plankton species even within replicates of similar $CO_2$ perturbation. Thus, small deviations in the initial

relative mass distribution of photoautotrophs, zooplankton, and detritus between replicate mesocosms can translate into some pronounced variability in measurements even under similar environmental conditions (e.g. Eggers et al., 2014).

Here we investigate data and their variability of replicate mesocosms during the PeECE-I experiment. For this we take a modelling approach to simulate environmental conditions and the predominant dynamics of nine individual mesoscosms as described in Engel et al. (2005) and in Delille et al. (2005). Joassin et al. (2011) presented a dynamical model to simulate the

mass flux of carbon (C), nitrogen (N), and of phosphorus (P) for the same PeECE-I experiment. Their model resolves growth





and losses of *E. huxleyi* together with interdependencies between bacteria, viruses, detritus, and dissolved organic matter (DOM). The model of Joassin et al. (2011) also features the exudation and coagulation process of dissolved polysaccharides (here referred to as dissolved combined carbohydrates, dCCHO) to form transparent exopolymer particles (TEP). In the study of Joassin et al. (2011) some emphasis is put on the enhanced mortality of *E. huxleyi* due to viral lysis and on the variable

stoichiometry (C:N ratio) of the particulate organic matter (POC:PON ratio). They did not attempt to resolve a dependency between calcification and $CO_2$ concentration and therefore restricted their simulations to one treatment with three replicate mesocosms that were exposed to present day $CO_2$ concentrations.

The focus of our model approach is different in that we distinguish between two phytoplankton functional types, calcifying algae (e.g. *E. Huxleyi*) and bulk non calcifying algae, e.g. an unresolved combination of picoplankton, dinoflagellates and

diatoms. We assume a $CO_2$ sensitivity for the ratio of calcification versus net carbon fixation (photosynthesis minus respiration), based on results from the meta-analysis of Findlay et al. (2011). In our data-model synthesis we concentrate on the initialisation (initial filling) of the mesocosms, with possible variations in the relative distribution of plankton and detritus resolved in our model. A data assimilation (DA) method is employed for the estimation of parameter values, which helps to disentangle and understand some of the differences and commonalities seen in observations, in particular in TA and PIC data, but also in

measurements of dissolved inorganic nitrogen (DIN) and DIC, chlorophyll *a*, as well as in particulate organic nitrogen (PON) and POC.

First we will briefly provide some background information about the experimental setup of PeECE-I, including irradiance, temperature and salinity, as these environmental factors enter our model simulations. This will be followed by a description of the model equations that include components of the optimality-based approach to simulate algal growth, using parameterisa-

tions proposed by Pahlow et al. (2013). Thereafter, the DA method for parameter estimation will be explained. Specific details of the model and of the DA method are given in the Appendix. Ensembles of three distinct model solutions will be presented together with their mass flux estimates of C and N. We will discuss the problem of identifying initial conditions in combination with important model parameters. We will also address the problem of resolving the variability observed in the accumulation of PIC and how this variability is related to the expression of the $CO_2$ effect introduced to the model.

## 2   Material and methods

For our analysis we consider the setup and data of the PeECE-I experiment that was conducted at the Marine Biological Field Station (Raunefjorden, 60.3°N, 5.2°E) of the University of Bergen, Norway between 31 May and 25 June 2001 (Engel et al., 2005; Delille et al., 2005). The objective of this study was to investigate OA effects on marine calcifying algae (coccolithophores) captured in polyethylene bags of enclosed water volumes (mesocosms) and perturbed by different levels

of $CO_2$ concentrations. A dynamical plankton ecosystem model is used for simulations of N and carbon C flux within each mesososm. We apply a DA method to identify best estimates of model parameter values together with initial conditions for model simulations.



## 2.1 Data

Nine mesocosms of 2 m diameter and 11 m$^3$ volume were filled with unfiltered, post-bloom, nutrient depleted water from the fjord. After the filling of the mesocosms nutrients were added so that all mesocosms had similar initial nutrient concentrations, approximately 15 mmol m$^{-3}$ of nitrate together with nitrite and 0.5 mmol m$^{-3}$ of phosphate. Like the nutrients, the initial

total alkalinity (TA) in all nine bags were approximately 2146 mmol m$^{-3}$ approximately (or if normalised to unit mass $\approx$ 2200 $\mu$mol kg$^{-1}$). The bags were covered with air-tight tents of tetra-fluroethylene foil that allowed 95% of photosynthetically active radiation (PAR) to pass through. The mesocosm bags were subject to three different levels of perturbation of partial pressure of CO$_2$: a) mesocosms 1-3, referred to as M1, M2, and M3, were exposed to similarly high DIC levels (initial DIC = 2119 mmol m$^{-3}$ , 2119 mmol m$^{-3}$ , 2122 mmol m$^{-3}$) with 700 ppmV of initial $p$CO$_2$, b) M4, M5, and M6 started from DIC = 2048

mmol m$^{-3}$ , 2056 mmol m$^{-3}$ , 2040 mmol m$^{-3}$ with a corresponding $p$CO$_2$=370 ppmV, and treatment c) with initial DIC = 1919 mmol m$^{-3}$, 1929 m$^{-3}$, 1927 m$^{-3}$ with 180 ppmV $p$CO$_2$ in mesocosms M7, M8, and M9. Thus, data from three replicate mesocosms are available for each of the three CO$_2$ treatments. For each mesocosm the partial pressure of atmospheric CO$_2$ above the surfaces was largely controlled by a continuous injection of gas with a treatment specific, individually prescribed CO$_2$ content. Because there was an open space between surface of mesocosms and the tents, we assumed the $p$CO$_2$ in the

air above the mesocosms' surfaces to be a mixture of 90% of the pertubed $p$CO$_2$ inside a mesocosm and 10% of the actual atmospheric $p$CO$_2$ (340 ppm) in all replicates.

Daily samples were collected and measured over a period of 23 days. For every mesocosm temperature and salinity data were interpolated to hourly values for direct use as environmental input for model simulations (Fig. 1). Hourly photosynthetic available radiation (PAR) data were derived from meteorological global irradiance measurements of the Geophysical institute

at Bergen (Skartveit et al., 2001). Figure (1) shows that temperature increased by approximately 3 Degree Celsius during the experiment and variations between the different mesocosms remained small. Small but noticeable differences exist between mesocosms with respect to salinity. In all mesocosms a gradual decrease in salinity was observed, from S=31.3 to approximately S=30.8. The PAR data exhibit variations on an hourly scale, due to changes in cloud cover.

## 2.2 Modelling approach

For model simulations we assume that all mesocosms are homogeneously mixed, as we neglect an explicit representation of vertical turbulent mixing (0D-model approach). The applied model equations describe mass exchange rates of N and C between compartments of 1) dissolved inorganic nitrogen and carbon (DIN and DIC), 2) N and C biomass of coccolithophores and other phytoplankton (CoccoN and CoccoC , PhyN and PhyC), 3) zooplankton (ZooN and ZooC), and 4) detritus (DetN and DetC), and 5) labile dissolved organic N and C (DON and DOC), Fig. (2). As due to the design of the PeECE-I experiment our model

includes some additional features. The first is that we distinguish between bulk phytoplankton biomass and the presence of calcifying algae, coccolithophores like *E. huxleyi*. Furthermore, we consider an explicit representation of dissolved combined carbohydrates (dCCHO) that act as precursors for transparent exopolymer particles (TEPC), similar to Schartau et al. (2007) and Joassin et al. (2011). Finally, our model has to resolve changes in TA along with DIC so that we can also derive pH values



and the corresponding partial pressure of $CO_2$ ($pCO_2$). We neither resolve viral infections nor bacterial biomass explicitly, as done in Joassin et al. (2011). Microbial activity is implicitly considered by parameterisations of hydrolysis and remineralisation; both processes are assumed to be independent of changes in bacteria biomass. Instead, hydrolysis and remineralisation rates are calculated as being proportional to substrate availability only. Likewise, any effects by viral lysis remain unspecified and are

an integral part of a single total mortality that is assigned to phytoplankton and coccolithophores. In the following, the general model equations of mass flux of C and N are described as sources and sinks, inducing changes in the mass concentration of the respective state variables.

### 2.2.1 Photoautotrophs

In our model we distinguish between calcifying and non-calcifying photoautotrophs, coccolithophores (Cocco) and other bulk

phytoplankton (Phy). Respective net photoautotrophic growth rates ($\mu_{\text{cocco/phy}}$) are described as rates of gross carbon fixation ($V^C$) minus some corresponding sum of respiration costs ($r_C$) due to the synthesis of chlorphyll $a$, nutrient assimilation, and maintenance: $\mu_{\text{cocco/phy}} = V^C - r_C$. The proportions of $V^C$ and $r_C$ are determined by optimal resource allocation while energetic trade-offs are imposed, as described in Pahlow et al. (2008). These physiological equations of optimal allocation have been shown to be well applicable for a series of different conditions (e.g. including diazotrophy) and scales (e.g. Smith et al.,

2011; Pahlow et al., 2013; Arteaga et al., 2014; Fernández-Castro et al., 2016). Here we neglect diazotrophy as well as the effect of phosphorus availability on nitrogen uptake and thus on algal growth. From the data we could not infer any phosphorus limitation of growth prior to nitrogen depletion and we assume that cellular nitrogen (N) directly limits the net growth rate of photoautrophs ($\mu_{\text{cocco/phy}}$). Nitrogen is generally necessary for enzymatic reactions. According to the model approach of Pahlow and Oschlies (2009), the major metabolic pathways within the algae are regulated by the resources allocated to produce

these enzymes. Thus, key processes like photosynthesis, chlorophyll $a$ synthesis and net carbon fixation become affected by internal resource allocation. The model maximise the photoautotrophic growth rates by optimising the allocation of resources to nutrient acquisition sites and to the light harvesting complex (LHC). The detailed equations for resource allocation are given in the Appendix (A).

    The resource allocation depends on the cellular nitrogen-to-carbon (N:C) ratio, expressed by the cell quota ($Q^N$). $Q^N$ is the

cellular N biomass normalised to carbon/energy units. The availability of resources that can be allocated is estimated by the relative difference between $Q^N$ from and a subsistence quota ($Q_0$). $Q_0$ is the minimum N:C ratio required for a photoautotrophic cell to survive. As $Q^N$ approaches $Q_0$ less resources can be allocated (e.g. to the LHC) and algal growth becomes limited. Light dependence of photoautrophic growth in the model is quantified by the degree of light saturation ($S_I$), which depends on the amount of chloroplasts that have been build up in the cell (expressing the status of the LHC) (Eq. A.9 in the Appendix A). An

optimal allocation of nutrients to specific cellular sites (or cell compartments) is thus determined by a trade-off between three fractions: a) a fraction that is allocated to the nutrient acquisition complex ($f_V$), b) a fraction attached to structural proteins (expressed as $Q_s/Q^N$), and c) a remaining fraction ($1 - f_V - Q_s/Q$) that can be allocated to the LHC and thus promotes the synthesis of chlorophyll $a$ (Pahlow et al., 2013). An optimal allocation factor ($f_V^0$) for nutrient uptake is derived by maximising net growth rate with respect to nutrient uptake and thus $f_V$ (Eq. A.3 in Appendix A). Under nutrient depleted conditions,



some higher growth rate of a algal cell can be maintained by increasing $f_V^0$ to the cost of resources that can be assigned to the light-harvesting complex (referred to as $f_{LHC}^0$; the optimal allocation factor for LHC). In consequence, the mobilisation of resources (N in this study) for nutrient acquisition (induced by an increase of $f_V^0$) reduces the rate of chlorophyll $a$ synthesis. Vice-versa for light-limited conditions, growth rate of a cell is optimised by investing more resources to LHC of a cell, which

enhances the rate of chlorophyll $a$ synthesis. This is achieved for low values of $f_V^0$.

**Biomass concentrations of photoautotrophs**: The biomass buildup (net growth) of photoautotrophs depends on the amount of N and C assimilated by the algae minus losses because of aggregation, grazing by zooplankton and because of exudation or leakage of organic matter. The sources minus sinks (sms) terms of the photoautotrophs' biomass are:

**sms of photoautotroph biomass** $=$ C and N uptake $-$ exudation/leakage $-$ aggregation $-$ grazing

The corresponding differential equations of C and N biomass for phytoplankton and coccolithophores are:

$$\frac{d}{dt}\text{PhyC} = (\mu_{phy} - C_{fact} \cdot \gamma_N) \cdot \text{PhyC} - \frac{A_{phy}}{Q_{phy}^N} - \frac{G_{phy}}{Q_{phy}^N} \tag{1}$$

$$\frac{d}{dt}\text{CoccoC} = (\mu_{cocco} - C_{fact} \cdot \gamma_N) \cdot \text{CoccoC} - \frac{A_{cocco}}{Q_{cocco}^N} - \frac{G_{cocco}}{Q_{cocco}^N} \tag{2}$$

$$\frac{d}{dt}\text{PhyN} = V_{phy}^N \cdot \text{PhyC} - \gamma_N \cdot \text{PhyN} - A_{phy} - G_{phy} \tag{3}$$

$$\frac{d}{dt}\text{CoccoN} = V_{cocco}^N \cdot \text{CoccoC} - \gamma_N \cdot \text{CoccoN} - A_{cocco} - G_{cocco} \tag{4}$$

A description of auxiliary variables is given in Table (1). We stress that the parameterisations in Eqs. (1 and 2) are identical for both photoautrophic groups (coccolithophores and non-calcifying algae), but some of the corresponding optimised parameter values may turn out to be different between the two. The detailed equations of aggregation ($A_{cocco/phy}$) and grazing

($G_{cocco/phy}$) are given in the Appendix (A).

**Chlorophyll $a$ concentrations**: The synthesis of chlorophyll $a$ (Chl) is represented by an optimal trade-off between photosynthesis and respiratory costs in the chloroplast of a cell. The synthesis rate depends on the amount of net carbon fixed inside chloroplasts and change in $\theta$ due to changes in $Q^N$ which depends on degree of light saturation ($S_I$). As for biomass, the

parameterisations for chlorophyll $a$ are identical for the calcifying and non-calcifying phytoplankton in our model:

**sms of chlorophyll** $a =$ synthesis of chlorophyll $a -$ aggregation $-$ grazing

The respective differential equations for chlorophyll $a$ of non-calcifying phytoplankton (with subscripts $phy$) and coccolithophores ($cocco$) are:

$$\frac{d}{dt}\text{Chl}_{cocco/phy} = (\mu_{cocco/phy} + \frac{\dot{\theta}_{cocco/phy}}{\theta_{cocco/phy}}) \cdot \text{Chl}_{cocco/phy} - A_{cocco/phy} \cdot \theta_{cocco/phy}^N - G_{cocco/phy} \cdot \theta_{cocco/phy}^N \tag{5}$$



where $\theta_{cocco/phy}$ are the cellular Chl:C ratios of phytoplankton and coccolithophores respectively, given in units mg Chl (mmol C)$^{-1}$. Similarly, $\theta^N_{cocco/phy}$ are the respective cellular Chl:N ratios in units mg Chl (mmol N)$^{-1}$ (Table 1). The terms $\dot{\theta}_{cocco/phy}$ are the time derivatives of $\theta_{cocco/phy}$. The regulation of $\theta_{phy}$ and $\theta_{cocco}$ upon on the buildup and limitation of chlorophyll $a$ is determined by optimality-based criteria. The corresponding detailed equations are listed in the Appendix (A).

**Formation of particulate inorganic carbon (PIC)**: The process of calcification in our model depends on the amount of energy provided through photosynthesis and is simply expressed by a ratio of PIC formation per carbon fixed ($f_{PIC}$, Eq. A.20). The differential equation of PIC describes a net accumulation rate (formation minus dissolution) and no explicit distinctions can be made with respect to how PIC becomes eventually distributed between algal biomass, detritus or zooplankton:

**sms of particulate inorganic carbon** $=$ calcification by coccolithophores $-$ dissolution of coccoliths (calcite)

written as a single differential equation:

$$\frac{d}{dt}\mathrm{PIC} = (f_{CO_2} \cdot f_{PIC} \cdot \mu_{cocco}) \cdot \mathrm{CoccoC} - \tau_{dissol} \cdot \mathrm{PIC} \tag{6}$$

where $\tau_{dissol}$ is the dissolution rate of PIC, given in units d$^{-1}$. A reference rate of PIC formation under nutrient replete and light saturated conditions is prescribed as a molar ratio of $f_{PIC} = 0.5$ mol PIC formed per mol C assimilated into organic matter, Eq.
(A.20). The molar ratio ($f_{PIC}$) is assumed to increase when the fraction of resources allocated to the light harvesting complex (LHC) of a cell ($f0^{\mathrm{LHC}}_{cocco}$) decreases. According to our model approach the process of calcification can be interpreted as an additional pathway for dissipating excess energy (Barcelos e Ramos et al., 2012), as is the case under high light conditions when chlorophyll $a$ synthesis rates diminish (induced by a reduction of $f0^{\mathrm{LHC}}_{cocco}$). On the one hand, PIC formation becomes enhanced under high light conditions, while less resources become allocated to LHC. On the other hand, calcification is reduced
or ceases under conditions of low or no light. Under nutrient depleted conditions, when more resources become allocated to nutrient uptake sites rather than to LHC, the rate of calcification per net carbon fixation also increases. For low (nutrient limited) growth rates under saturated (or high) light conditions the parameterisation $f_{PIC}$ can yield maxima in the calcite-to-C$_{organic}$ ratio (of the calcifying algae) that may reach values of 2 and slightly above. The function $f_{CO_2}$ in Eq. (6) has no dimension and it simulates the effect of varying CO$_2$ concentrations on $f_{PIC}$. It is derived from results of a regression model of a meta-analysis
described in Findlay et al. (2011), Eq. (A.21) in the Appendix (A).

### 2.2.2 Zooplankton

The grazing losses of the photoautotrophs are resolved with an explicit representation of zooplankton biomass. With our grazing approach (Holling type III) no distinctions are made between micro- and mesozooplankton or between different feeding types. Changes in zooplankton biomass are subject to a mortality ($M_{zoo}$; e.g. losses to higher trophic levels). Other loss terms
represent respiratory costs ($r_{zoo}$) as well as excretion ($\gamma_{zoo}$), e.g. of urea. Zooplankton restore C and N towards a constant N:C ratio ($Q^{zoo}_{const}$) of 0.19. The restoring time ($\tau$) in our model is equal to one day. It mimics an increase in respiration ($r_{zoo}$) if N:C ratio falls below $Q^{zoo}_{const}$ and an increase in excretion ($\gamma^N_{zoo}$) if N:C is above $Q^{zoo}_{const}$. The buildup of zooplankton biomass





depends on the total prey concentrations (phytoplankton and coccolithophores):

**sms of zooplankton biomass** $=$ grazing on phytoplankton $+$ grazing on coccolithophore

$-$ respiration (or excretion) $-$ mortality

The respective differential equations are:

$$\frac{d}{dt}\text{ZooC} = \frac{G_{phy}}{Q_{phy}^N} + \frac{G_{cocco}}{Q_{cocco}^N} - r_{zoo} - \frac{M_{zoo}}{Q_{zoo}} \tag{7}$$

$$\frac{d}{dt}\text{ZooN} = G_{phy} + G_{cocco} - \gamma_{zoo}^N - M_{zoo} \tag{8}$$

The grazing function as well as detailed formulations of zooplankton respiration and excretion are given in the Appendix (A).

### 2.2.3 Detritus

Detritus comprises a variety of components with particles of different sizes and sinking rates (Fasham et al., 1990). The detritus resolved by our model simply combines dead plankton biomass and fecal pellets. Sources of detrital C,- and N mass are given in terms of phytoplankton aggregation and mortality of zooplankton. Aggregation is parameterised with quadratic loss terms of the photoautotrophs. These aggregation equations resolve interactions between two types of particles (small cells of photoautotrophs and large aggregates of detritus): a) aggregation of cells of photoautrophs and b) aggregation of small photoautotrophs with larger detritus, see details in the Appendix (A). The two particle-type approach allows a trade-off between accuracy of estimated mass flux and the resolution of particle size (Ruiz et al., 2002). We assume that hydrolysis is temperature dependent and that it is responsible for the degradation of detritus, acting as a source for (labile) $L$DON and $L$DOC. The equations of detrital C and N can thus be described as:

**sms of detritus** $=$ aggregation of phytoplankton $+$ aggregation of coccolithophore

$+$ zooplankton mortality $-$ hydrolysis

The corresponding differential equations are:

$$\frac{d}{dt}\text{DetC} = \frac{A_{phy}}{Q_{phy}^N} + \frac{A_{cocco}}{Q_{cocco}^N} + \frac{M_{zoo}}{Q_{zoo}} - \omega_{det} \cdot T_f \cdot \text{DetC} \tag{9}$$

$$\frac{d}{dt}\text{DetN} = A_{phy} + A_{cocco} + M_{zoo} - \omega_{det} \cdot T_f \cdot \text{DetN} \tag{10}$$

### 2.2.4 Dissolved inorganic compounds (DIN, DIC) and total alkalinity (TA)

**Dissolved inorganic nitrogen (DIN)**: The DIN pool represents the total concentration of nitrate, nitrite and ammonium. Nitrogen utilisation by phytoplankton and coccolithophores is a sink of DIN, whereas heterotrophic excretion and remineralisation



of $LDON$ are the major sources:

**sms of dissolved inorganic nitrogen** $= -$ N uptake by phytoplankton $-$ N uptake by coccolithophores

$+$ excretion by heterotrophs $+$ remineralisation

The nitrogen uptake ($V_{cocco/phy}^{N}$) is carbon-specific and is therefore given as a rate of N utilisation per carbon, in units mol N (mol C) $^{-1}$ d$^{-1}$ (Pahlow and Oschlies, 2009):

$$\frac{d}{dt}\text{DIN} = -(V_{phy}^{N} \cdot \text{PhyC} + V_{cocco}^{N} \cdot \text{CoccoC}) + \gamma_{zoo}^{N} + \rho \cdot T_f \cdot LDON \tag{11}$$

The sources of DIN are calculated from zooplankton excretion ($\gamma_{zoo}^{N}$) and the remineralisation of $LDON$.

**Dissolved inorganic carbon (DIC)**: The DIC pool combines $CO_2$, bicarbonate and carbonate. The primary sinks of DIC are net carbon fixation to support photoautotrophic growth ($\mu_{cocco/phy}$) and calcification of coccolithophores. We do not differentiate between the utilisation of $CO_2$ and bicarbonate for algal growth and calcification. Note that net carbon fixation ($\mu_{cocco/phy}$) in our model becomes slightly negative in the absence of light (dark respiration of the photoautotrophs). Total heterotrophic respiration acts as major DIC source and is expressed by zooplankton respiration and by the remineralisation of dissolved organic carbon ($LDOC$ + dCCHO):

**sms of dissolved inorganic carbon** $= -$ net C uptake by phytoplankton $-$ net C uptake by coccolithophores

$-$ calcification $+$ dissolution of PIC $+$ zooplankton respiration

$+$ remineralisation $+$ gas exchange

$$\frac{d}{dt}\text{DIC} = -\mu_{phy} \cdot \text{PhyC} - (1 + f_{CO_2} \cdot f_{pic}) \cdot \mu_{cocco} \cdot \text{CoccoC} + \tau_{dissol} \cdot \text{PIC} + r_{zoo} + \rho \cdot T_f \cdot (LDOC + \text{dCCHO}) + \text{F}_{\text{DIC}} \tag{12}$$

Calculations of air-sea gas exchange ($\text{F}_{\text{DIC}}$) within mesocosms are based on original carbonate chemistry code provided by the Ocean Carbon-Cycle Model Intercomparison Project (Orr, 1999). The original code was refined to include an accelerated iteration scheme for pH and $p\text{CO}_2$ calculations (Christoph Völker, personal communication), as already applied in Schartau et al. (2007).

**Total alkalinity (TA)**: Temporal changes in TA in our model are due to the process of PIC precipitation and dissolution of calcite plates produced by the calcifying algae. We follow the nutrient-H$^+$ compensation principle (Wolf-Gladrow et al., 2007) and TA variations induced by DIN uptake and the remineralisation of $LDON$ compounds are also accounted for. Furthermore, a fixed stoichiometric N:P ratio of 16 is assumed, in order to simulate accompanied TA responses to the utilisation and



remineralisation of phosphorus. The latter have only a minor effect on TA.

$$\textbf{sms of total alkalinity} = \text{N and P uptake by phytoplankton} + \text{N and P uptake by coccolithophores}$$
$$- \text{calcification by coccolithophores} + \text{dissolution of calcite}$$
$$- \text{remineralisation of dissolved organic N and P}$$

$$\frac{d}{dt}\text{TA} = (1 + 1/16) \cdot \left( \frac{V_{phy}^N}{Q_{phy}^N} \cdot \text{PhyN} + \frac{V_{cocco}^N}{Q_{cocco}^N} \cdot \text{CoccoN} \right) - 2 \cdot (f_{\text{CO}_2} \cdot f_{\text{PIC}} \cdot \mu_{cocco} \cdot \text{CoccoC} - \tau_{dissol} \cdot \text{PIC})$$
$$- (1 + 1/16) \cdot \rho \cdot T_f \cdot L\text{DON} \tag{13}$$

Measured values of DIN, TA, and DIC on day one of the experiment were taken as initial conditions for respective mesocosms.

### 2.2.5 Dissolved labile organic matter and transparent exoplymer particles

Dissolved organic matter (DOM) is produced by exudation of the photoautotrophs and by hydrolysis of detrital matter. The DOM is subject to remineralisation, being the source of DIN and DIC. The applied model distinguishes between dissolved combined carbohydrates (dCCHO) and a residual fraction of labile dissolved organic carbon and nitrogen ($L$DOC and $L$DON). This distinction is made because only dCCHO are simulated to act as precursors for the formation of transparent exopolymer particles (TEP). In our model the DOM's primary source is freshly exuded and leaked organic matter from photoautotrophs.

An addtional source of DOM is due to degradation of detrital matter (hydrolysis and microbial exudation) in response to bacterial activity. The fraction of exudates that enter the dCCHO pool may vary between expontential growth phase and during periods of nutrient limited growth, described as two modes of exudation in Schartau et al. (2007). We therefore introduced a parameterisation ($f_{\text{dCCHO}}^{cocco/phy}$, Eq. A.22) that simulates such shift in quality of the exudates, depending on the respective cell quota of the coccolithophores and of the other phytoplankton ($Q_{cocco/phy}^N$).

Remineralisation and microbial respiration are respective sinks of $L$DOC and $L$DON. The equations for labile DOC and DON are described as follows:

$$\textbf{sms of labile dissolved organic matter} = \text{exudation by photoautotrophs}$$
$$+ \text{hydrolysis/degradation of detritus} + \text{hydrolysis/degradation of gels}$$
$$- \text{remineralisation/respiration of dissolved organic matter}$$

$$\frac{d}{dt}L\text{DOC} = C_{fact} \cdot \gamma_N \cdot \left[ (1 - f_{\text{dCCHO}}^{phy}) \cdot \text{PhyC} + (1 - f_{\text{dCCHO}}^{cocco}) \cdot \text{CoccoC} \right] + \omega_{det} \cdot T_f \cdot \text{DetC}$$
$$+ \omega_{gel} \cdot T_f \cdot \text{TEPC} - \rho \cdot T_f \cdot L\text{DOC} \tag{14}$$

$$\frac{d}{dt}L\text{DON} = \gamma_N \cdot (\text{PhyN} + \text{CoccoN}) + \omega_{det} \cdot T_f \cdot \text{DetN} - \rho \cdot T_f \cdot L\text{DON} \tag{15}$$




**Dissolved combined carbohydrates (dCCHO)**: By introducing dCCHO we can account for an additional sink of DOC other than microbial degradation, which is the physical-chemical transformation of dissolved to particulate matter, here resolved as the coagulation of dCCHO to form TEP carbon (TEPC). This transformation is parameterised as aggregation process, as proposed in Engel et al. (2004b) and effectually applied in Schartau et al. (2007) and in Joassin et al. (2011):

**sms of dissolved combined carbohydrates** $=$ exudation $-$ coagulation of dCCHO

$$- \text{ aggregation of dCCHO with TEP} - \text{remineralisation of dCCHO}$$

$$\frac{d}{dt}\text{dCCHO} = C_{fact} \cdot \gamma_N \cdot \left[ f_{\text{dCCHO}}^{phy} \cdot \text{PhyC} + f_{\text{dCCHO}}^{cocco} \cdot \text{CoccoC} \right] - \phi_{\text{dCCHO}} \cdot \text{dCCHO}^2 - \phi_{\text{TEP}} \cdot \text{dCCHO} \cdot \text{TEPC}$$

$$- \rho \cdot T_f \cdot \text{dCCHO} \tag{16}$$

**Transparent exopolymer particles (TEP)**: The carbon content of TEP (TEPC) is explicitly resolved because it can be a significant constituent of POC measurements (Verdugo et al., 2004). This consideration is important for our data-model synthesis, in particular because it affects the stoichiometric C:N ratio of particulate organic matter. The sink terms of dCCHO, described before, are the only sources for TEPC in our model approach. The degradation of TEPC is parameterised similar to the hydrolysis of detritus:

**sms of transparent exopolymer particles (TEPC)** $=$ coagulation of dCCHO $+$ aggregation of dCCHO with TEP

$$- \text{ degradation}$$

$$\frac{d}{dt}\text{TEPC} = \phi_{\text{dCCHO}} \cdot \text{dCCHO}^2 + \phi_{\text{TEP}} \cdot \text{dCCHO} \cdot \text{TEPC} - \omega_{gel} \cdot T_f \cdot \text{TEPC} \tag{17}$$

The derivation of respective values for $\phi_{\text{dCCHO}}$ and $\phi_{\text{TEP}}$ are given in the Appendix (A), Eqs. (A.23 and A.24).

### 2.2.6 Model parameters and initial conditions

Out of 33 model parameters, 26 parameters are fixed and remaining 7 parameters (4 initial condition parameters ($f_{cocco}$, $f_{zoo}$, $f_{det}$. $PON_0$) and 3 ecological parameters ($\alpha_{phy}$, $\alpha_{cocco}$, $Q_0$) enter the optimisation procedure that will be described in Sect. (2.3). Initial condition values for some of the state variables in the model are computed by initial condition parameters, given in fractions. The initial biomass during the start of the experiments, specified by $PON_0$, is distributed between living and non-living biomass, which is determined by the parameter of the initial detritus fraction ($f_{det}$). The living biomass is further distributed between photoautotrophs and zooplankton, specified by the initial zooplankton fraction parameter ($f_{zoo}$). Finally, the remaining relative distribution of photoautotrophic biomass is set by $f_{cocco}$. For example, a value of $f_{cocco} = 1$ would mean





that all photoautotrophic biomass is associated with the presence of coccolithophores exclusively.

$$PON_0 = \quad\quad DetN_0 + ZooN_0 + CoccoN_0 + PhyN_0 \tag{18}$$

with the individual fractions:

$$DetN_0 = \quad\quad f_{det} \cdot PON_0 \tag{19}$$

$$ZooN_0 = \quad\quad f_{zoo} \cdot (PON_0 - DetN_0) \tag{20}$$

$$CoccoN_0 \quad\quad f_{cocco} \cdot (PON_0 - DetN_0 - ZooN_0) \tag{21}$$

$$PhyN_0 = \quad\quad (1 - f_{cocco}) \cdot (PON_0 - DetN_0 - ZooN_0) \tag{22}$$

For initial zooplankton, coccolithophore and phytoplankton biomass we apply a constant C:N ratio of 6.625. We assume a higher C:N ratio (= $2 \cdot 6.625$) only for initial detritus. Initial condition of PIC, DIC, and TA are taken from the data for respective mesocosm, whereas we assume same small fixed values as initial conditions for DON, DOC, dCCHO and TEPC for all mesocosms.

## 2.3 Design of data assimilation (DA) approach

A peculiarity of the PeECE-I experiment is that high and low changes in total alkalinity (TA) were found in all three $CO_2$ treatments, in response to differences in calcification (Delille et al., 2005). Because the three distinct patterns in calcification are attributable to all three treatments means that a factor other than the $CO_2$ perturbations induced variations between the individual mesocosms. For all other observations no such clear pattern could be identified. We designed our DA approach according to this finding and therefore investigate three possible situations (model solutions) that differ in their TA response: low, medium and high calcification (referred to as LC, MC, and HC respectively). Thus, for each of these three situations we can consider data from three mesocosms that were subject to three different $CO_2$ levels (initial 700 ppmV, 370 ppmV, and 180 ppmV). By adapting the same nomenclature as in Engel et al. (2005) and in Delille et al. (2005), we assign the mesocosms M1, M6, and M8 to those with low calcification rates (highest TA), M2, M5, and M7 to the ones with medium calcification and finally M3, M4, and M9 to mesocosms with high calcification rates (lowest TA).

### 2.3.1 Definition of cost function (data-model misfit)

In our DA approach we consider data from the three cases (LC, MC, and HC) separately, but we make identical statistical assumptions. A maximum likelihood (ML) estimator is applied, meaning that no explicit prior information is considered for the estimation of parameter values. For each calcification case we assume observational residual errors to be additive, and daily standard errors ($\boldsymbol{\sigma}$) could be derived from observations of three mesocosms. Correlations between measurements were computed based on data of all nine mesocosms. Eventually, we use three similar cost functions but with data ($\boldsymbol{y}$) and covariances ($\mathbf{R}$) from the respective three mesocosms of each case. Seven different types of data are considered (dimension of $\boldsymbol{y}$ is $N_y$ = 7). These daily data ($\boldsymbol{y}_i$) are available for a period of $N_t$ =23 days, with subscript $i$ indicating the day when measurements were made. The elements of the parameter vector of interest ($\Theta$) are those parameters listed in Table (2), including the initial





value of $PON_0$ and initial condition parameters that further specify how $PON_0$ is distributed between detritus, zooplankton, coccolithophores and the remaining photoautotrophs.

The ML estimation we assume that the maximum posterior probability of the parameter estimates given the data, $p(\Theta|\boldsymbol{y})$, is proportional to the maximum of the likelihood $p(\boldsymbol{y}|\Theta)$, which is the probability of explaining the data given a set of values
assigned to each parameter (to each element of $\Theta$):

$$p(\Theta|\boldsymbol{y}) \propto p(\boldsymbol{y}|\Theta) = \text{constant} \cdot \exp[-\frac{1}{2}\sum_{i=1}^{N_t} \boldsymbol{d}_i^T \mathbf{R}_i^{-1} \boldsymbol{d}_i] \propto \exp[-\frac{1}{2} J(\Theta)] \tag{23}$$

with $\boldsymbol{d}_i = \boldsymbol{y}_i - H_i(\boldsymbol{x})$ being the data-model residual at date $i$, which is the difference between the vector of observations ($\boldsymbol{y}_i$) and the corresponding counterparts of model results $H_i(\boldsymbol{x})$ (the elements of vector $\boldsymbol{x}$ are the model's state variables). The ML estimate of parameter values can be found by actually identifying the minimum of the exponent of $p(\boldsymbol{y}|\Theta)$ of Eq. (23), since
the constant term is independent of $\Theta$. We thus compute and minimise the following cost function $J(\Theta)$:

$$J(\Theta) = \sum_{i=1}^{N_t} (\boldsymbol{y}_i - H_i(\boldsymbol{x}))^T \mathbf{R}_i^{-1} (\boldsymbol{y}_i - H_i(\boldsymbol{x})) \tag{24}$$

Eventually we not only wish to identify the minimum of $J(\Theta)$ that corresponds with one best estimate of parameter values ($\widehat{\Theta}$) but also confine a credible region of parameter estimates. This credible region tells us how reliable the parameter estimates are (yielding lower and upper credibility limits) and resolves correlations (collinearities) between the parameters.

The observation vector ($\boldsymbol{y}_i$) contains daily means of three mesocosms of the following measurements: 1) dissolved inorganic carbon (DIC, mmol m$^{-3}$), 2) dissolved inorganic nitrogen (DIN) (nitrate + nitrite, mmol m$^{-3}$), 3) chlorophyll $a$ (Chl $a$, mg m$^{-3}$), 4) particulate organic nitrogen (PON, mmol m$^{-3}$), 5) particular ogranic carbon (POC, mmol m$^{-3}$), 6) particulate inorganic carbon (PIC, mmol m$^{-3}$), 7) total alkalinity (TA, mmol m$^{-3}$):

$$\boldsymbol{y}_i - H_i(\boldsymbol{x}) = (\text{data} - \text{model results}) = \begin{pmatrix} \text{DIC}_i \\ (\text{NO}_3^- + \text{NO}_2^-)_i \\ \text{Chl } a_i \\ \text{PON}_i \\ \text{POC}_i \\ \text{PIC}_i \\ \text{TA}_i \end{pmatrix} - \begin{pmatrix} \text{DIC}_i \\ \text{DIN}_i \\ (\text{CHL}^{cocco} + \text{CHL}^{phy})_i \\ (\text{PhyN} + \text{CoccoN} + \text{ZooN} + \text{DetN})_i \\ (\text{PhyC} + \text{CoccoC} + \text{ZooC} + \text{DetC} + \text{TEPC})_i \\ \text{PIC}_i \\ \text{TA}_i \end{pmatrix} \tag{25}$$

Like the data vector $\boldsymbol{y}_i$, the vector $H_i(\boldsymbol{x})$ represents mean values of three simulated mesocosms for each calcification case (LC, MC, and HC). It combines results of model states: C and N biomass concentrations of the photoautotrophs (PhyN & PhyC and CoccoN & CoccoC), of zooplankton (ZooN & ZooC), of detritus (DetN & DetC), and carbon concentration of transparent exopolymers particles (TEPC).



For the cases LC, MC, and HC we calculated daily residual standard errors ($\sigma_i$) based on the measurements. The $\sigma_i$ are the diagonal elements of a matrix $\mathbf{S}_i$ whose off-diagonal elements are zero, see Eq. (B.3) in Appendix (B). The time-varying covariance matrices $\mathbf{R}_i$ are constructed with $\mathbf{S}_i$ together with some correlation matrices ($\mathbf{C}_{(\boldsymbol{y})}$). The matrices $\mathbf{C}_{(\boldsymbol{y})}$ have been derived from data for two distinct periods: 1) the exponential growth phase, and 2) the post-bloom period.

$$\mathbf{R}_i = \mathbf{S}_i \cdot \mathbf{C}_{(\boldsymbol{y})} \cdot \mathbf{S}_i \tag{26}$$

Equation (26) is applied because correlations between observations can change from pre-bloom period to post-bloom period. For example, PON and DIC are strongly negatively correlated during the exponential growth phase but become weakly positively correlated during the post-bloom period, when both, DIC and PON, decrease. The correlation matrices, $\mathbf{C}_{(\boldsymbol{y})}$, for the two respective periods are given in the Appendix (B).

### 2.3.2 Parameter optimisation procedure

The entire optimisation procedure of each (LC, MC, and HC) case is subject to five consecutive analysis steps:

1) adjustment of parameters while considering published typical values $\longrightarrow$ specify model solution that is in qualitative (visual) good agreement with observations of the medium calcification MC case

2) application of *simulated annealing* algorithm (SANN), see (Bélisle, 1992), to effectively scan and minimise the seven-dimensional manifold ($\Theta, J(\Theta)$), while avoiding to get trapped into local minima of $J(\Theta)$ $\longrightarrow$ obtain global estimate of $\Theta$

3) local refinement of the parameter estimate, using the *Broyden-Fletcher-Goldfarb-Shannon* (BFGS) algorithm (Broyden, 1970; Fletcher, 1970; Goldfarb, 1970; Shannon, 1970) $\longrightarrow$ identify maximum likelihood estimate that corresponds with the global minimum ($\widehat{\Theta},, J(\widehat{\Theta})$)

4) calculation of the inverse of second derivatives of $J(\Theta)$ with respect to every parameter ($\mathcal{H}_{jj} = \partial^2 J / \partial \Theta_j^2$ at $\widehat{\Theta}$, which is a point-wise approximation of the diagonal elements of a Hessian matrix $\mathcal{H}$) $\longrightarrow$ derive marginal errors (standard errors, $\sqrt{\mathcal{H}_{jj}^{-1}}$) of the estimated parameter values

5) application of a *Monte Carlo Markov Chain* (MCMC) method, using the marginal error information of 4) to confine credible range of optimal parameter values $\longrightarrow$ derive posterior confidence limits of parameter estimates and collinearities (correlations) between parameter estimates.

For steps 2, 3, and 5 the R package FME is applied, as coded and described by Soetaert and Petzoldt (2010). The plankton ecosystem model was coded and compiled as shared library in FORTRAN so that we can apply a FORTRAN-R wrapper function. This wrapper allows us to take advantage of fast numerical Euler forward integrations of the model equations while, at the same time, we can benefit from the R platform and its freely available packages. The cost function $J(\Theta)$ is evaluated in R. The MCMC method employed here is based on the *Adaptive Metropolis-Hastings* (AMH) algorithm (Haario et al., 2001),





which is also available with the R package FME. The AMH algorithm generates a new parameter vector ($\Theta^*$) by perturbing the original vector $\Theta$, inferred from a "proposal" distribution (Metropolis et al., 1953). The marginal error information (Step 4) is required for the proposal (Gaussian) distribution in the AMH algorithm to generate $\Theta^*$. We approximated the diagonal elements of the Hessian with finite central differences, as described in e.g. Matear (1995), Kidston et al. (2011), and in Kreus

and Schartau (2015). To do so we imposed an incremental step size of 1% variation to the respective parameter values.

## 3 Results

### 3.1 Parameter estimates for specific mesocosms with low, medium, and high calcification

The same seven model parameters were optimised for all three calcification cases (LC, MC, and HC) independently, using data from respective mesocosms. With our DA approach we can thus specify commonalities and differences between model

solutions for mesocosms with LC, MC and HC. Table (3) lists all ML estimates, which correspond with the best model solutions obtained with the MCMC method. Note that the estimates given in Table (3) are not the ensemble means of MCMC results. Collinearities are expressed by the correlation coefficients of two parameter combinations, which we have also calculated from results of the MCMC method (Table 4).

Credible intervals for each parameter were derived from nonparametric probability densities of the MCMC estimates. The

corresponding posterior probabilities distributions are the cumulative sums of these nonparametric probabilty densities (CDF) in Fig. (3). The steeper the CDF increase the narrower the 95% credible interval of the parameter estimate. According to the width of credible intervals we find uncertainty ranges of initial conditions parameters $f_{det}$, $f_{zoo}$ and $PON_0$ to be generally small for all three cases of calcification respectively. The initial condition parameters are best constrained for the solution of medium calcification (MC). The parameter $f_{cocco}$ shows the largest uncertainty for the HC case. A large fraction ($\approx 90$ %)

of initial biomass comprises of detrital matter in all three solutions. Table (5) shows mean concentration values of $PON_0$, $DetN_0$, $ZooN_0$, $CoccoN_0$ and $PhyN_0$ along with standard errors according to respective MCMC estimates. Initial zooplankton concentration is highest in HC solutions. Thus more photoautotrophic biomass is lost due to grazing by zooplankton and less by aggregation in model solutions for HC, which is reflected by the negative correlation between initial condition parameters $f_{zoo}$ and $f_{det}$.

For those parameters that do not specify the initial conditions we hoped to find all credible intervals to overlap, which would have suggested insignificant differences between the estimates. A single set of values of these parameters could then be unambiguously used for simulations of all nine mesocosms, independent of how the values of the initial conditions turned out to be. This is not the case, as can be seen in Fig. (4) and in the correlation coefficients (Table 4). Estimates of the subsistence quota ($Q_0$) are lower for the mesocosms with high and medium calcification rates. Apparently, lower $Q_0$ and higher $\alpha_{cocco}$ values

are required to build-up high coccolithophores biomass in mesocosms with high calcification rates as initial coccolithophores concentration is low and grazing pressure is high.





## 3.2 Data-model comparison

The posterior distribution resolved by the MCMC method yields an ensemble of optimal parameter vectors for each calcification case. The variational range of these ensemble of parameter estimates (Figs. 3 and 4) induce ensembles of model trajectories (model results) that are statistically indistinguishable (or equivalent). Based on these posterior ensemble parameter estimates

of all three calcification solutions we find a general good agreement between model results and the data, (Fig. 5).

The ensembles reflect uncertainty ranges in model solutions, which correspond nicely with most of the variability in observations. Almost the entire range of variability in TA is recovered with our three distinct solutions of calcification. The observed variability in POC is captured with the optimal ensemble model solutions. Only few maximum values seen in POC data remain unresolved, likely because we have optimised parameters that hardly introduce changes in the solution of TEPC concentrations.

The simulated POC concentration constitutes TEPC and some part of the variability seen remains unresolved. From preceding analyses we learned that one parameter, namely $CN_{fact}$, is effectively determining the maximum in simulated TEPC concentration. The parameter $CN_{fact}$ did not enter the optimisation and we assume exudation and leakage rates to equal for C and for N. By allowing $CN_{fact}$ to vary between 1 and 2 we again see considerable variations in TEPC maxima, eventually pushing simulated POC values to the observed maxima (Fig. 6).

The model solutions exhibit some faster increase in the accumulation of PON during the exponential growth phase, in spite the fact that DIN data are matched well. Although this systematic model offset is pronounced, it does not correspond with any similar model bias in POC. Another general offset can be seen for simulated Chl $a$ concentrations during the post-bloom period. Our model shows sharp draw down in Chl $a$ in all three solutions (HC, MC and LC) during the post-bloom period, whereas observed Chl $a$ values are more variable.

### 3.2.1 Variations in calcification in response to growth conditions

According to our model approach we resolve changes in the rate of calcification relative to the carbon that is assimilated for growth of the coccolithophores. Figure (7) shows for the period of nutrient repletion much smaller variations in the molar calcification-to-C-assimilation ratio than under nutrient depleted growth conditions. All ensembles of model solutions (LC, MC, and HC) reveal a similar behaviour, with variations in $\Delta$PIC : $\Delta$C between 0.5 and 2 for growth rates between 0 and 0.3

d$^{-1}$. These variations depend on the light-acclimation state (e.g. $\theta_{cocco}$), fluctuations in irradiance and on cell quota ($Q^N_{cocco}$). The variations in $\Delta$PIC : $\Delta$C during nutrient depleted period can be attributed to fluctuations in carbon assimilation due to production of TEPC (Fig. 6).

### 3.2.2 Distinctions between model results of low, medium, and high calcification (LC, MC, and HC)

Optimised model results of low calcification (LC) yield the highest TA values of all mesocosms that are in accordance with the

TA data. DIN concentrations are well resolved by the model and variations of the ensemble DIN simulations are similarly low as in observations. The previously mentioned biases in PON and Chl $a$ are most conspicuous in this LC ensemble of optimal model results. Variability in the POC data of the LC mesocosms is not captured by the model ensemble. Model solutions are tightly



constrained, likely because we did not consider $CN_{fact}$ for optimisation, as explained before. But simulation results (solid lines in Fig. 8) match the POC mean of the three mesocosms. For PIC we also find a good agreement between model ensemble results and data. However, a noticeable potential bias exists for the PIC response in the high $CO_2$ treatment (M1), where model results overestimate PIC data during the maximum bloom period and shortly after nutrient depletion. This overestimation is

more pronounced in mesocosm with high $CO_2$ treatment. The LC ensemble successfully reproduces amplitude of Chl $a$ peak seen in data, this is also the case in the solutions of MC and HC mesocosms. Like for our LC solutions, DIN is well reproduced by optimal MC (medium calcification) solutions. Chl $a$ shows some faster increase in comparison to observations between day 4 and 11 in MC solutions (Fig. 9).

Simulated POC fits data the best in MC compared to other two solutions, however the model slightly overestimates observed
POC during bloom period. PIC and TA in MC solution appear well constrained and fits well to the mean of PIC and TA data from M2, M5 and M7. For the MC case, the model mode ensemble do not capture the observed full variability in PIC and TA . As in LC, PIC concentrations are noticeably overestimated by the model during the post-bloom period in the high $CO_2$ treatment (M2). DIC model values fit nicely to the data in the MC solution.

DIN (dissolved inorganic nitrate) is well resolved in the HC (high calcification) solutions (Fig. 10) , as in the MC and
LC. Simulated Chl $a$ also fits well to observations. HC solutions yield largest variability in DIC, TA and PIC amongst all optimised solutions, which we mainly attribute to the large uncertainty ranges of the model parameters $f_{cocco}$ and $\alpha_{cocco}$. HC solutions show sharp drawdown in DIC during the bloom period compared to other two solutions (MC and LC). This can be attributed to enhanced calcification activity due to high growth rates of coccolithophores in HC during the bloom period. Again, model overestimates observed PIC values (M3) under high $CO_2$ conditions shortly after the maximum of bloom. PON
is best reproduced in this HC case in comparison to MC and LC. Although model's HC solutions reproduce manage the entire variability in observed PIC, the corresponding best fits (to M3, M4, M9) underestimate PIC data.

### 3.2.3   Integrated flux estimates of carbon and nitrogen (C and N budgets of mesocosms)

The ensemble model solutions for LC and HC constitute two extremes and we therefore concentrate on the C and N budgets of these two cases. Carbon and N flux estimates were computed as the integral over the entire 23 days period. Figure (11)
show mean C and N flux estimates and their standard errors of the LC solution are given, distinguishing between low and high $CO_2$ treatments. Figure (12) shows the corresponding flux estimates for the HC solution. First of all, from these flux estimates we learn that the $CO_2$ effect introduced to the model, following Findlay et al. (2011), induces deviations in C flux that are much smaller than the variational range in model results, as reflected by the respective standard errors. Furthermore, differences between LC and HC flux estimates are larger than the responses to changes in $CO_2$ conditions. In both cases (LC
and HC) most inorganic carbon and nitrogen (DIC and DIN) are utilised by non-calcifiers ($\approx 56$ % in case of HC and $\approx 64$ % in the LC solution), despite the differences between LC and HC with respect to the initial biomass fractions. Generally, more carbon fixation (with C:N uptake ratio of $168:10 \approx 17$) occurs in the HC than in the LC mesocosms (C:N uptake ratio $\approx 13$). Carbon flux estimates show, carbon fixation in mesocosm with high $CO_2$ treatment is slightly higher than in the mesocosm with low $CO_2$ treatment. Flux budgets show that non-calcifiers clearly dominate in mesocosms with low calcification rates, and



in HC mesocosms coccolithophores and bulk phytoplankton biomasses are comparable (Figs. 11 and 12). Although grazing, in general, is high in HC mesocosms (Table 5), there is a trend of higher grazing pressure on bulk phytoplankton than on coccolithophores. This is shown by N flux estimates, where zooplankton gain nearly 57 % of their total biomass through grazing on non-calcifiers in HC and LC. Our results show, regardless of biomass, coccolithophores are always less vulnerable

to grazing than bulk phytoplankton. A noticeable difference between high and low calcification model ensembles is in terms of mortality of zooplankton. Higher mortality is seen in HC solutions. Since the carbon fixation in HC is high, exudation and leakage rates are also higher. Accordingly, TEPC production is enhanced in HC solutions. Unlike estimates of C flux, the N fluxes in HC and LC ensembles are similar, e.g. aggregation losses of phytoplankton and of coccolithophores are $3 \pm 0.4$ and $2 \pm 0.4$ mmol N m$^{-3}$ in HC, and $3.4 \pm 2 \cdot 10^{-3}$ and $1.5 \pm 2 \cdot 10^{-3}$ mmol N m$^{-3}$ in LC respectively. Similarly, flux estimates

in mesocosms with high calcification and low calcification rates show almost same rates of DIN utilisation, excretion, exudation and remineralisation.

## 4   Discussion

The DA approach applied in this study was designed to resolve differences in TA and thus in calcification while variations in other data (e.g. DIN, PON, and POC) should also be explained with our model. We distinguished between mesocosms

with high, medium and low calcification rates (HC, MC, and LC) and their respective data were used to come up with optimal estimates of initial conditions and of some important physiological model parameters. Ideally, we would have identified similar optimal values of the physiological parameters and would have obtained different estimates of the initial conditions for all three cases, HC, MC and LC respectively. However, our results reflect a more complex picture and our optimised values for the initial conditions also depend on the best estimates for the model parameters. The initial conditions could not be constrained

independently and model solutions of the HC case do not automatically imply a higher initial abundance of coccolithophores relative to the other, non-calcifying, phytoplankton. Likewise, the LC solution does not require a lower initial biomass of calcifying algae. Instead of differences in relative species abundance, the initial physiological conditioning, e.g. acclimation states of the algae, seems relevant as well, which is in the end reflected in the estimates of the physiological parameters $Q_0$, $\alpha_{cocco}$, and $\alpha_{phy}$). An alternative DA approach would be to optimise the physiological model parameters ($Q_0$, $\alpha_{cocco}$, and

$\alpha_{phy}$) together with the initial conditions ($PON_0$, $f_{det}$, $f_{zoo}$, and $f_{cocco}$) for mesocosms of one calcification case in a first step, e.g. the MC case (using data of mesocosms M2, M5, M7). In a second step we could have fixed the optimised physiological model parameters $Q_0$, $\alpha_{cocco}$, and $\alpha_{phy}$ (as identified with data of e.g. the MC case) and would have then estimated only the initial condition parameters for the other mesocosms, e.g. low and high calcification (LC and HC). This alternative approach does work (not shown), but we learned that we may then put too much confidence into those estimates of $Q_0$, $\alpha_{cocco}$, and

$\alpha_{phy}$ obtained first, e.g. estimates for the MC mesocosms. It can even obscure the fact that collinearities exist between some initial condition estimates and the other model parameters. Furthermore, with such alternative approach we could end up with different estimates of the initial conditions, if we would have started with data of either the HC or LC mesocosms first instead.




The design of our DA approach is more challenging but it is better suited to disclose major uncertainties and collinearities in estimating initial conditions together with model parameters of algal growth.

## 4.1 Uncertainty ranges in parameter estimates and variability in model solutions

Large variations can be seen in the data of PIC, reflecting the variability measured in TA. Since optimal ensembles of model
solutions were derived for three distinct cases of calcification (LC, MC and HC), we automatically capture most of the observed variability in PIC with our simulations. The spread of the ensemble solutions for TA and PIC is smaller in each of the three cases relative to the observed total range. This means that the respective uncertainties in our parameter estimates are small enough to obtain three distinctive ensembles of model solutions. However, it appears from our results that it is not possible to identify optimal values of the initial condition parameter $f_{cocco}$ independently from estimates of the other physiological model
parameters. This situation is aggravating but not unusual (Schartau et al., 2016). For instance, in a sensitivity study with a regional marine ecosystem model Gibson and Spitz (2011) stressed that collinearities exist between initial conditions and the values assigned to the biological parameters.

The posterior uncertainties in the estimates of the subsistence quota, $Q_0$, are rather small if compared with the uncertainty ranges of the other parameter estimates. Likewise, parameter estimates of the initial condition parameters $PON_0$, $f_{det}$, and
$f_{zoo}$ are fairly confined. Therefore, the variational range that we see in our model solutions is mainly induced by uncertainties in estimates of the photosynthesis parameters $\alpha_{cocco}$, $\alpha_{phy}$ and of $f_{cocco}$. The combination of these three parameters mainly determine the spread in model solutions with respect to the amount of C-fixation and also calcification. This also explains why the ensemble model solutions exhibit only small variations in DIN and PON concentrations and thus in our N-flux estimates.

Variability in POC is much more pronounced than in PON. All three model solutions show a steep increase in POC:PON
ratio as soon as algal growth becomes nutrient limited (Fig. 13). The variability seen in the POC:PON ratio is thus mainly due to a temporal variation in $Q^N$ (N:C ratio of both photoautotrophs) and thus of the algal growth conditions. The temporal variations in $Q^N$ eventually disperse into zooplankton biomass and detritus, inducing elevations of their respective C:N ratios. Another contribution to the elevation of POC:PON ratios is related to changes in POC because it constitutes concentrations of TEPC, which is explicitly resolved in our model. Although TEPC and dCCHO data were not assimilated in our DA approach, our
respective simulated concentrations compare well with those of Engel et al. (2004a). TEPC concentrations were also simulated in the study of Joassin et al. (2011) for the same mesocosm experiment. Our model results of dCCHO and TEPC do not show an abrupt increase in concentrations shortly after the beginning of the experiment as found in the simulations of Joassin et al. (2011). In their model simulations a short-term accumulation of TEPC occurred between day 5 and day 10 of the experiment. Such early increase in TEPC concentration is mainly suppressed in our model solutions, because we considered qualitative
changes in the exudation of DOC. The fraction of dCCHO exudates increases under nutrient limited growth conditions. In our model results the major increase in TEPC concentrations therefore happens only shortly after the onset of nutrient depletion (around day 13), which leads to a better agreement with POC measurements during the post-bloom period.

Our results show a large variability in molar $\Delta$PIC : $\Delta$C-assimilation at low net growth rates ($\mu_{cocco}$) under nutrient limited conditions (Fig. 7) in all three cases (HC, MC and LC). These variations are in the end translated in some variability seen





in the PIC:POC ratio. Variability in PIC:POC are discussed in Engel et al. (2014), where they collected and analysed data of diverse experiments and documented an increase (up to fourfolds) in values of cellular PIC:POC at relative growth rate (RGR) $\approx 0.2$ d$^{-1}$ and below in various $CO_2$ treatments. The reason for a sharp increase in molar $\Delta$PIC:$\Delta$C-assimilation ratio at low growth rates in our model is because of a down regulation of light harvesting complex (LHC). Such model behaviour is in

agreement with the interpretation of Barcelos e Ramos et al. (2012), who describe calcification as a process into which the coccolithophores can channel excess energy. In order to maximise (optimise) growth rate under nutrient depleted and high light conditions, the model allocates more resources and energy to support nutrient acquisition than to the LHC (indicated by low $f0_{cocco}^{LHC}$ values). Since $\Delta$PIC:$\Delta$C-assimilation is inversely related to $f0_{cocco}^{LHC}$ in our model, an increase in calcification (relative to C-fixation) is obtained at low growth rates. The maximum of $\Delta$PIC:$\Delta$C-assimilation ratio in our simulations are in

accordance with those found in Barcelos e Ramos et al. (2010).

### 4.1.1 Differences between high and low calcification solutions (HC and LC)

The optimised model solutions for HC and LC reveal significant differences in the development of coccolithophore biomass. As discussed before, these differences are not solely attributable to differences in the relative proportions of initial biomass concentrations. In fact, the optimisations yielded estimates that suggest fairly similar initial coccolithophore biomass con-

centrations between all nine mesocosms. Eggers et al. (2014) stressed that variations in initial plankton composition can be responsible for large differences in the responses observed on community level, thereby masking any possible $CO_2$ effect on photosynthesis or calcification. From a modelling perspective it is helpful to know about the initial individual mass contributions to $PON_0$, including details in the initial composition of the plankton. But the level of compositional detail remains unclear, since these variations in individual plankton composition will in the end always translate into some variational (uncer-

tainty) range in e.g. the initial photo-acclimation state, since our model approach only distinguishes between calcifiers and all other, non-calcifying, phytoplankton. These considerations were disregarded when we designed this study and we originally thought of the importance of the relative mass distributions between the state variables resolved by our model, while imposing fixed initial stoichiometric ratios (C:N and Chl$a$:N). It seems plausible to allow for some variations of the initial stoichiometric ratios as well.

For now we are interested in the question: what induces the different model solutions for LC and HC, in spite of similar initial conditions in the concentrations of coccolithophores and phytoplankton? First of all, we have some differences between the relative proportions of initial detrital, zooplankton, and photoautotrophic biomass (e.g. DetN:ZooN:(PhyN+CoccoN) = 80:10:1 for HC and 28:3:1 for LC). The difference between these ratios point towards net photoautotrophic growth rates that are higher in the LC case than in the HC case, since losses due to grazing and aggregation must be lower in the LC case. However, the

initial conditions in mesocosms of the LC case do not automatically yield model solutions of highest photoautotrophic growth. Instead we find overall reduced growth rates but some pronounced differences in the relative proportions of biomass between the coccolithophores and the non-calcifying phytoplankton (Fig. 14). The reason for these differences lies primarily in the relative differences between the estimates of the physiological parameters, with estimates of $\alpha_{cocco}$ being always smaller than of $\alpha_{phy}$. The photosynthetic efficiency of the coccolithophores remains clearly smaller (LC case) or can become similar (HC



case) relative to the other, non-calcifying, phytoplankton. Major differences between the LC and HC solutions can thus be attributed to higher $\alpha_{cocco}$ values (median $\alpha_{cocco}$ = 1.7 mol C (g Chl $a$)$^{-1}$ m$^2$ W$^{-1}$ d$^{-1}$) in HC posterior distribution compared to LC (median $\alpha_{cocco}$ = 1.4 mol C (g Chl $a$)$^{-1}$ m$^2$ W$^{-1}$ d$^{-1}$). The model solution is highly sensitive to the values assigned to the parameter $\alpha_{cocco}$, hence a difference of 0.3 mol C (g Chl $a$)$^{-1}$ m$^2$ W$^{-1}$ d$^{-1}$ can effectively determine the differences

in our simulations with respect to rates of carbon fixation and calcification. The build-up of comparable nitrogen biomass of coccolithophores and bulk phytoplankton in HC solutions are achieved with identical $Q_0$ values and only nuanced differences in values between $\alpha_{cocco}$ and $\alpha_{phy}$. In contrast, bulk phytoplankton (non-calcifiers) outcompete coccolithophores during the bloom period in the LC solutions (Fig. 14).

Differences in photosynthetic efficiency estimates for the LC and HC cases could possibly be invoked for two reasons: a)

because of unresolved differences in initial photo-acclimation states (e.g. different light-history during the filling period), since we assume identical initial Chl:N ($\theta^N_{cocco} = \theta^N_{phy}$) and N:C ($Q_{cocco} = Q_{phy}$) ratios for all nine mesocosms (and thus for LC, MC, and HC), or b) because of unresolved varying conditions in irradiance. To impose identical surface PAR forcing on all nine mesocosms might not be appropriate and the arrangement of neighbouring mesocosms may have caused some shading effects. From the available data and with our model approach it is not possible to resolve such varying conditions afterwards.

## 4.2   Model biases

Model biases and compensating effects are typically seen when applying DA methods (Bertino et al., 2003; Gregg, 2008). Our simulated PON concentrations show an early offset during the early growth period. This offset is most conspicuous in the solutions of the MC mesocosoms (Fig. 9). For the MC case the model yields optimised solutions with a build-up of coccolithophores biomass that is apparently too fast (Fig. 15). The estimates of $f_{cocco}$ turned out to be highest, if compared with the

estimates for the low and high calcification (LC and HC) model solutions. Furthermore, the range of credible values for $f_{cocco}$ is small (Fig. 3). The initial PON concentration ($PON_0$) is also highest for the MC case. Both estimates, of $f_{cocco}$ and of $PON_0$, lead to an inital biomass concentration of coccolithophores that is approximately three times higher than in the LC case and even six times the initial concentration of the model solutions for HC. This apparent overestimation of initial coccolithophore biomass concentration is responsible for the noticeable bias (temporal offset) in simulated PON concentrations.

With our model we do not distinguish between growth of picoplankton and the other non-calcifying phytoplankton during the initial bloom phase. The initial abundance of picoplankton (mainly *Micromonas spp.*) and their decline was observed during the early pre-bloom period of the PeECE-I experiment (Engel et al., 2005). This explains why our simulated Chl $a$ and PON concentrations are lower compared to observations between day 1 and day 4. Another discrepancy between simulated and observed Chl $a$ exists during the post-bloom period. We assume that this bias is mainly because we do not account for detrital

chlorophyll pigments (presumably of inactive or destroyed cells) in our model. Formation of detritus is associated with the aggregation of coccolithophores and of the other phytoplankton to form detritus (simulated as a transfer of algal biomass into detritus) in our model, and the fate of Chl $a$ within the detritus compartment remains unresolved. Once N and C biomass of the photoautotrophs are transformed to detritus, an associated flux of Chl $a$ is disregarded. An explicit consideration of the fate of Chl $a$ would likely improve model performance and some refinements in this respect are recommended for the future.





Results of our data-model synthesis also exhibit a small but distinctive bias in the calcification response to elevated CO2 levels. The distinctions we made with respect to mesocosms of LC, MC and HC helped us to identify such bias, which will be addressed in the following section.

### 4.3 Disentangling $CO_2$ effect from the observed variability in PIC

We considered a simple $CO_2$ relationship that mimics only OA effects on calcification. This dependency was adopted from the meta-analysis of Findlay et al. (2011). With this $CO_2$ dependence we can already capture differences in PIC formation. The $CO_2$ sensitivity that we introduced to our model is only effective with respect to the ratio of calcification versus C-fixation, thereby reducing the overall calcification rate under high $CO_2$ conditions. This effect turned out to be small compared to the total variability seen in PIC data. According to our model setup we do not consider any potential changes in vulnerability to
predation (or edibility) of the coccolithophores due to elevated $CO_2$. Likewise, any additional $CO_2$ effects, e.g. on the rate of aggregation, are not accounted for. Such effects remain unresolved and therefore the comparison of our budget calculations yield only small differences between high and low $CO_2$ levels, in particular with respect to nitrogen flux estimates. Thus, differences in C and N budgets between the two extreme calcification cases, LC and HC, are more pronounced than between different levels of $CO_2$. To resolve consecutive ecological effects in response to a reduction of the relative calcification rate we
would have needed explicit data, i.e. revealing differences in grazing and aggregation rates between the individual mesocosms. With the PON and POC data used in our DA approach it is not possible to distinguish between different coccolithophore loss terms like grazing and aggregation, since detritus and zooplankton are both constituents of the same PON and POC measurements.

    The advantage of resolving LC, MC and HC solutions separately is that for each case we can compare data with model
results of mesocosms individually, of low (glacial), medium (present), and high (future) $CO_2$ treatments. In other words, for every LC, MC, and HC case we resolve three mesocosms, of which each was subject to different $CO_2$ levels. This way we have separated differences between LC, MC, and HC from variations induced by a $CO_2$ effect. Doing so reveals PIC formation to be be systematically overestimated by the model for all mesocosms of the future treatment. The overestimation is only detectable during the period of bloom and few days after the bloom in all HC, MC and LC solutions (Figs. 8, 9 and 10). In contrast to
Delille et al. (2005), our results show an early onset of calcification in mesocosms of the high $CO_2$ treatment between day 10 and day 15. It indicates that the $CO_2$ effect introduced to our model is likely too weak. This becomes evident according to positive model-data residuals in PIC between day 13 and day 18 for those mesocosms with future treatment (Fig. 16). It is not evident for the glacial and present day $CO_2$ treatments, where the corresponding residuals do not show a systematic positive offset.

Figure (17) shows the total variability seen in PIC data together with the full variational range of all ensemble model solutions. In addition, we depict those ranges in simulated PIC that are solely due to the $CO_2$ effect, based on the two extreme calcification solutions (lowest and highest simulated PIC) and the best model solution (according to the lowest cost function values) for the MC mesocosms. If we compare the simulated $CO_2$ response signal on calcification with the total variability in





PIC (in Fig. 17) we find that the $CO_2$ effect remains small. This situation demonstrates the difficulty in isolating a distinctive $CO_2$ signal from the variability seen in PIC observations.

## 5  Conclusions

With our DA approach we could disentangle three distinctive ensembles of model solutions (LC, MC and HC) that represent

mesocosms with high, medium and low calcification rates. The full spread of model ensemble solutions reproduce most of the observed variability in calcification (PIC production). From this modelling study we infer that collinearities exist between estimates of initial conditions and physiological model parameters, in particular for the photosynthetic efficiencies $\alpha_{phy}$, $\alpha_{cocco}$ and the initial fraction of coccolithophores determined by $f_{cocco}$. Therefore, it is not possible to identify initial concentration of photoautotrophs independent of parameters responsible for phytoplankton growth in HC, MC and LC model solutions.

This inference justifies our DA approach of optimising model parameters (initial conditions and physiological parameters) separately for HC, MC and LC mesocosms.

By separating the model solutions for mesocosms with high, medium and low calcification rates we could better specify the $CO_2$ effect on PIC formation. For mesocosms exposed to high $CO_2$ levels (future treatments) we identified a systematic overestimation of calcification in our model and we conclude that the simulated $CO_2$ effect on PIC formation is too weak.

Furthermore, our results suggest that the variability seen in PIC and TA data might not be only due to differences in initial biomass composition but may also depend on the photo-acclimation states that prevailed within the phytoplankton community during the filling of the mesocosms. Overall, the results of our data-model synthesis document the need to resolve initial plankton composition, and even possibly trace details about physiological acclimation states, since these conditions may have greater impact on the plankton community dynamics that can mask physiological and ecological responses to $CO_2$ perturbations.

*Acknowledgements.* The development of the modelling framework for mesocosm simulations and data assimilation was supported by the large integrated project Surface Ocean Processes in the Anthropocene (SOPRAN, 03F0662A), funded by the German Federal Ministry of Education and Research (BMBF). This study is a contribution to the BMBF funded BIOACID (03F0728A) project. We gratefully acknowledge support by Markus Pahlow, who helped to refine equations in our model. We also like to acknowledge support given by Andreas Oschlies and by the GEOMAR data management team. We thank Sabine Mathesius for the compilation and inclusion of the forcing data into the

mesocosm modelling setup. We thank Yonss Jose and Hadi Bordbar for helpful and constructive comments.



## 6  Figures

**Figure 1.** Forcing variables for all nine mesocosms: The upper panel shows temperature, linearly interpolated to hourly values between daily observations. The middle panel displays hourly interpolated salinity values and the lower panel reveals the irradiance data with hourly temporal variations resolved.





**Figure 2.** Schematic representation of the model: boxes characterise individual compartments that are represented by one and more model state variable. The arrows represent key biogeochemical processes (named in red) between compartments. One compartment includes dissolved inorganic carbon and nitrogen (DIC and DIN). This comartment also embeds total alkalinity (TA). Biomass concentrations of photoautotrophs are resolved with respect to carbon and nitrogen explicitly (referred to as PhyC and CoccoC, and PhyN and CoccoN respectively). Variations in carbon and nitrogen biomass are also resolved for zooplankton (ZooC and ZooN) and for detritus (DetC and DetN). Dissolved combined carbohydrates (dCCHO) are distinguished from other labile dissolved organic matter, desribed as LDOC and LDON. Only dCCHO are assumed to act as precursor for the formation of transparent exopolymer particles, whose carbon content is explicitly resolved (TEPC). One compartment represent the formation and dissolution of particulate inorganic carbon (PIC), affecting DIC as well as TA.





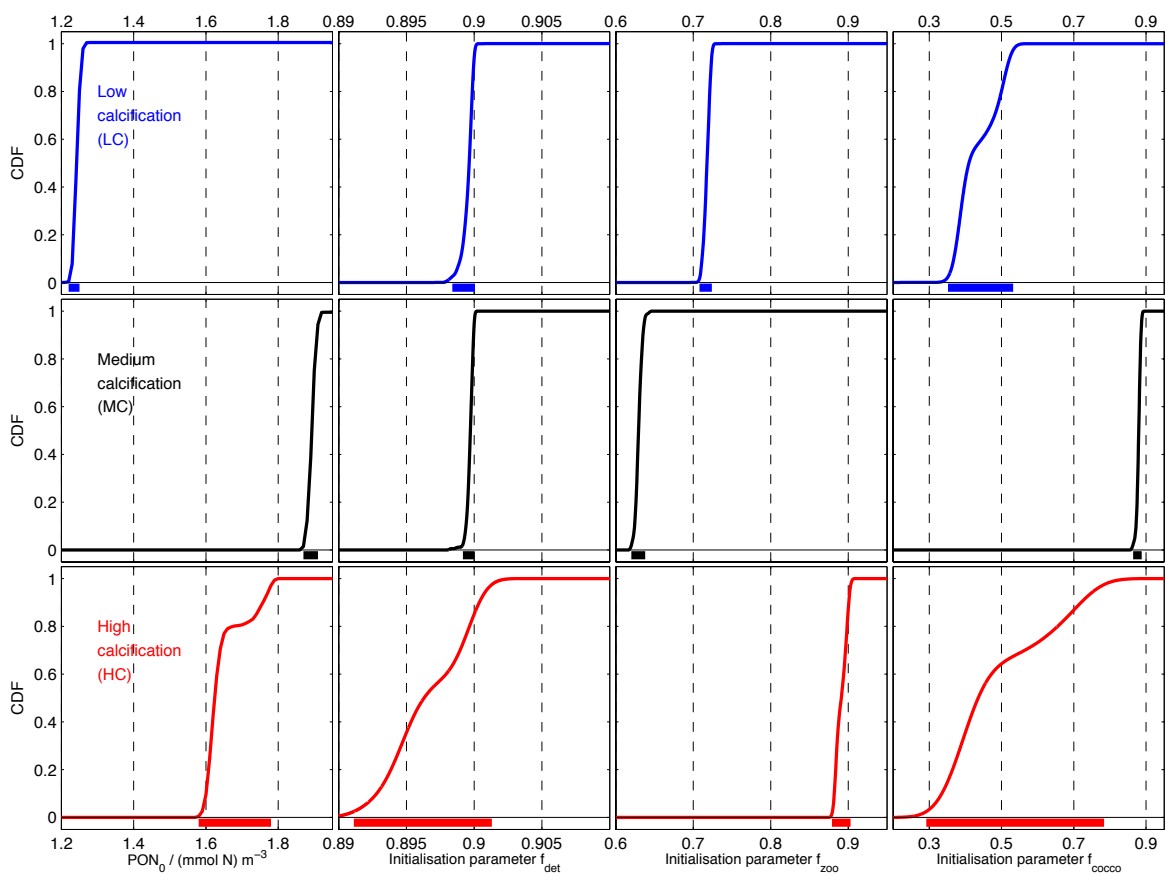

**Figure 3.** Probability distributions of the initial condition parameters: the cumulative sum of non-parametric probability densities (CDF) were derived from the posteriors of the Markov Chain Monte Carlo (MCMC) approach. The bars on the bottom of each panels show respective 95% credible (uncertainty) ranges of the parameter estimates.





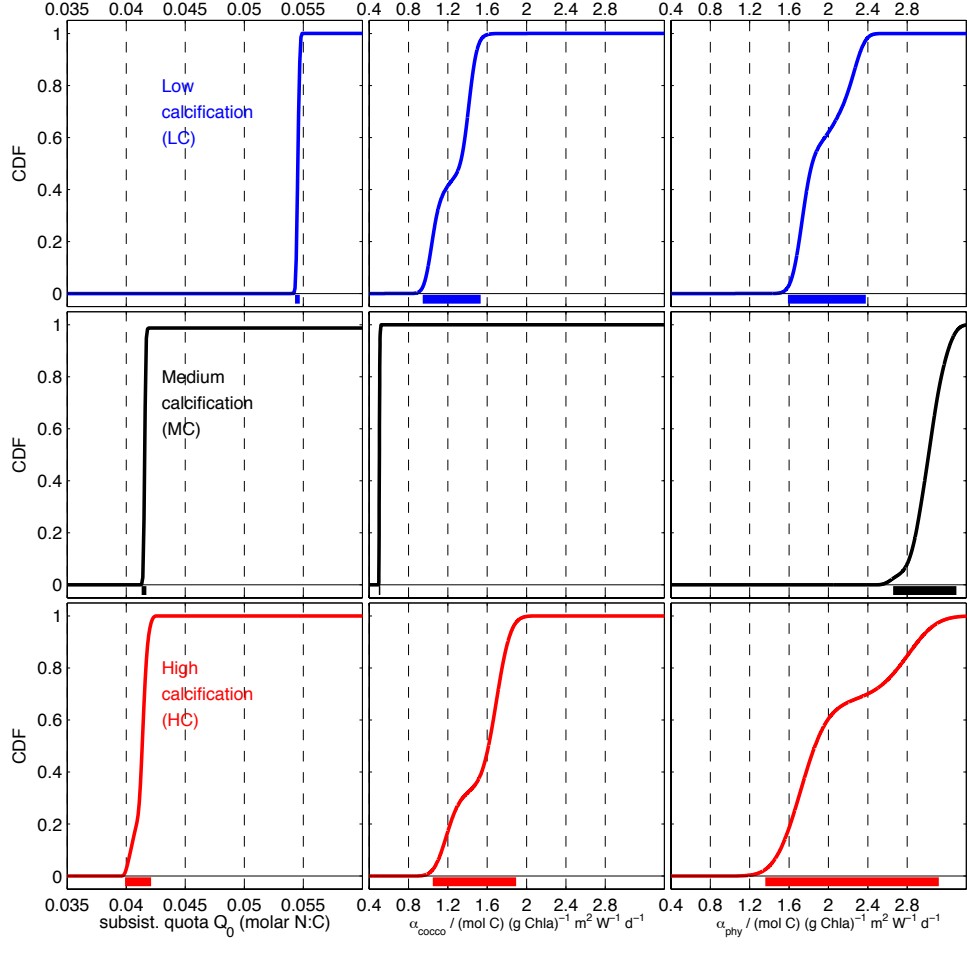

**Figure 4.** Probability distributions of the physiological model parameters: the cumulative sum of non-parametric probability densities (CDF) were derived from the posteriors of the Markov Chain Monte Carlo (MCMC) approach. The bars on the bottom of each panels show respective 95% credible (uncertainty) ranges of the parameter estimates.





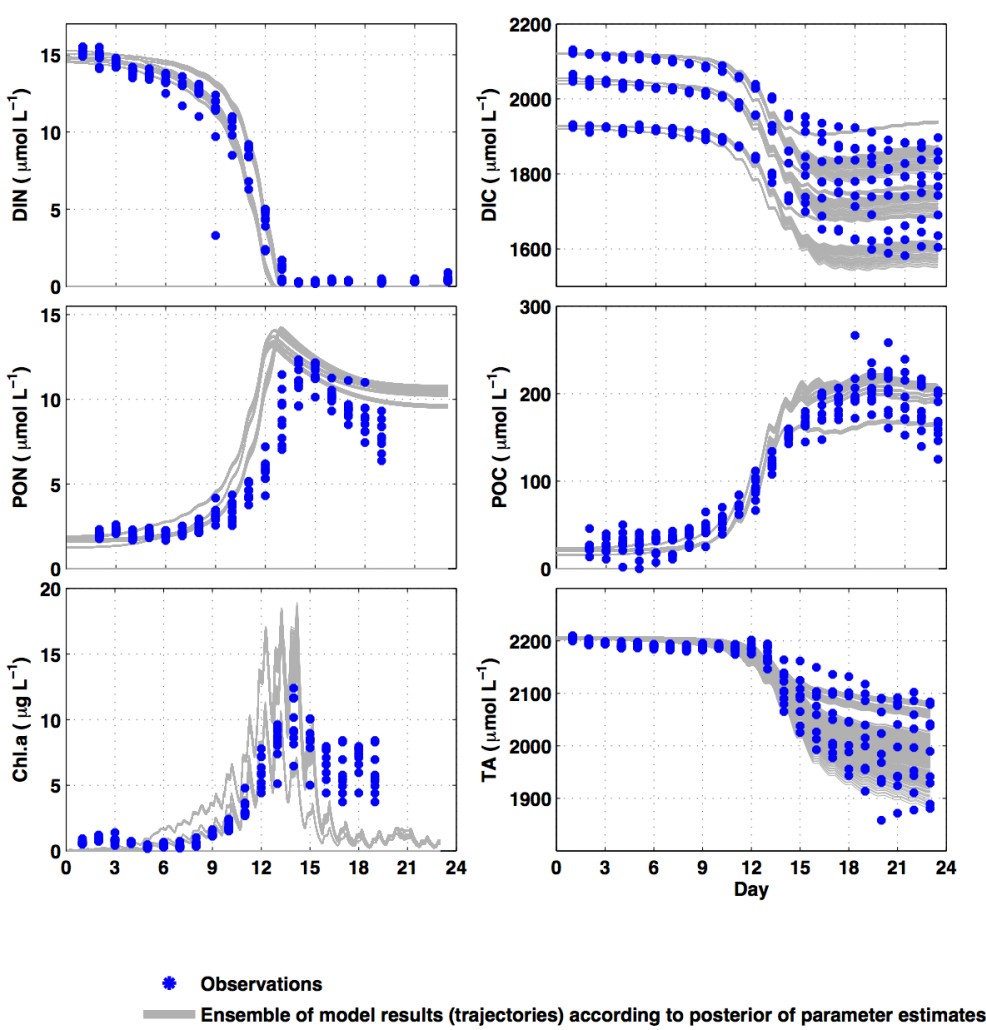

**Figure 5.** Full variational range of model outputs due to uncertainties in parameter estimates. Model ensembles of high, medium and low calcification solutions compared with observations.





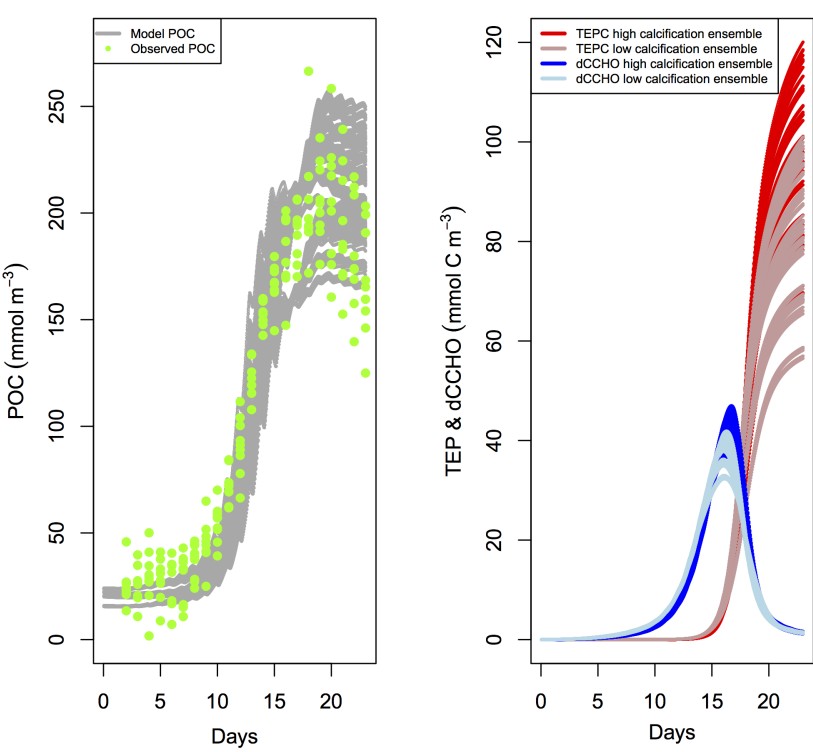

**Figure 6.** First panel shows variability in observed POC captured by varying model parameter $CN_{fact}$. Second panel shows TEP and dCCHO model outputs corresponding to variations in $CN_{fact}$.





**Figure 7.** Molar calcification-to-C-fixation ratio compared to net growth rate of coccos ($\mu_{cocco}$) in high, medium and low calcification solutions, while also considering variations in the model parameter $CN_{fact}$, with random values between 1 and 2.





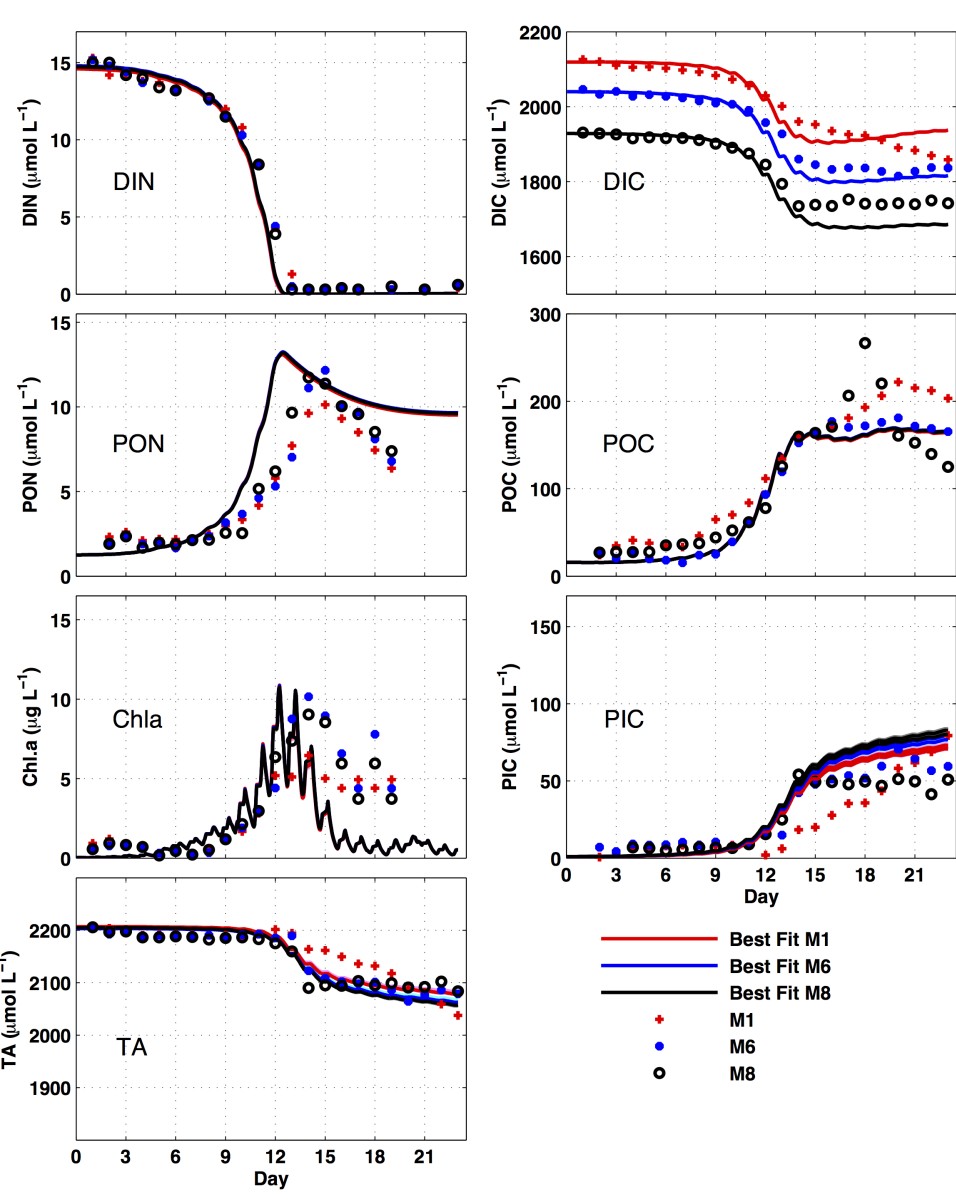

**Figure 8.** Low calcification solution. The coloured bands represent ensemble of model results according to *a posteriori* and symbols show observations.





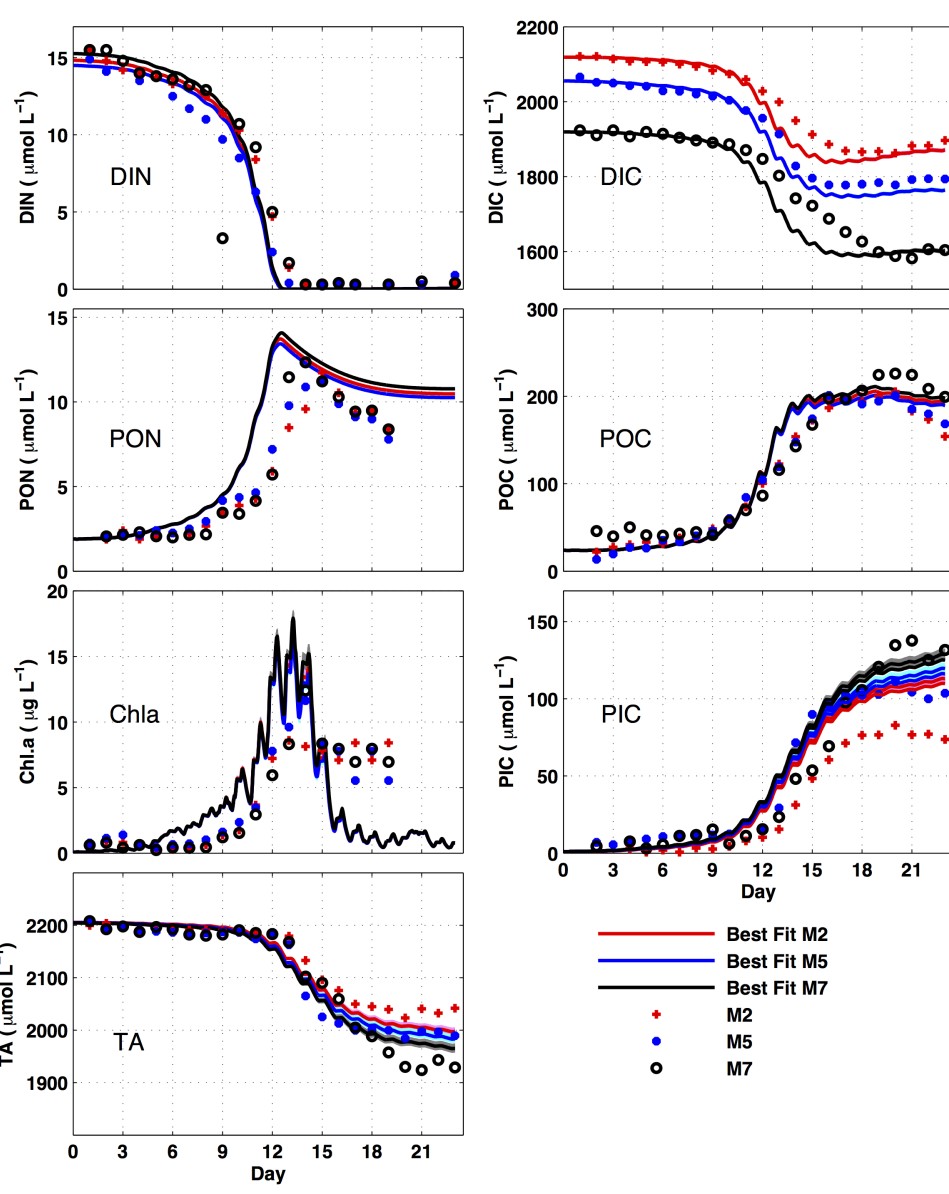

**Figure 9.** Medium calcification solution. The coloured bands represent ensemble of model results according to *a posteriori* and symbols show observations.





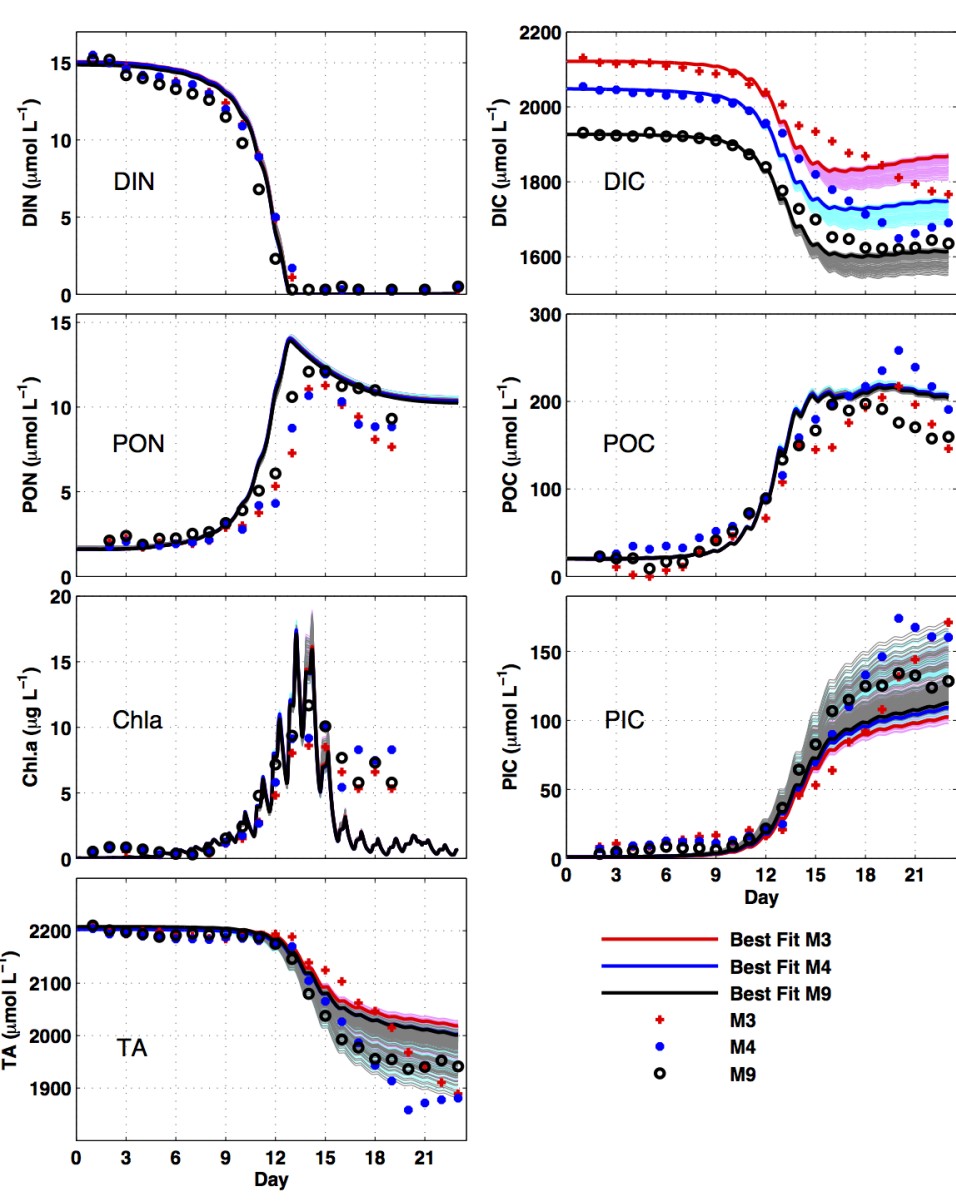

**Figure 10.** High calcification solution. The coloured bands represent ensemble of model results according to *a posteriori* and symbols show observations.





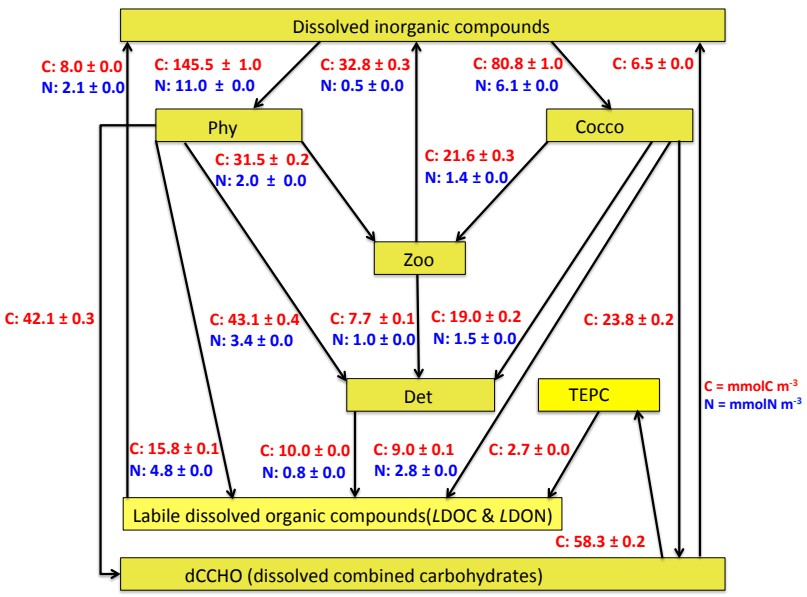

**(a) High CO$_2$**

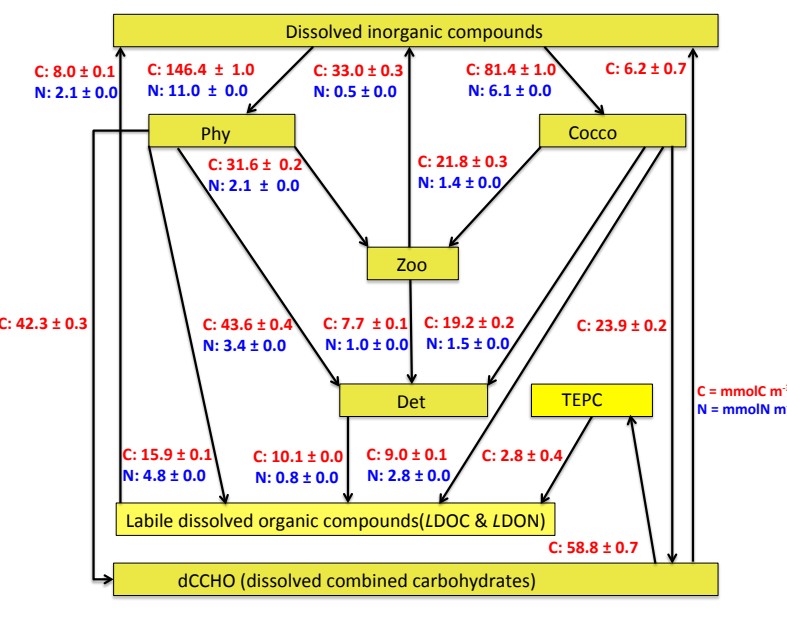

**(b) Low CO$_2$**

**Figure 11.** Carbon and nitrogen fluxes estimated by the model in mesocosms with low observed calcification but different CO$_2$ treatment, high (a) and low (b). All the arrows that point downwards show flux estimates from the respective compartment on the right hand side, whereas arrows pointing upwards show values on the left hand side.





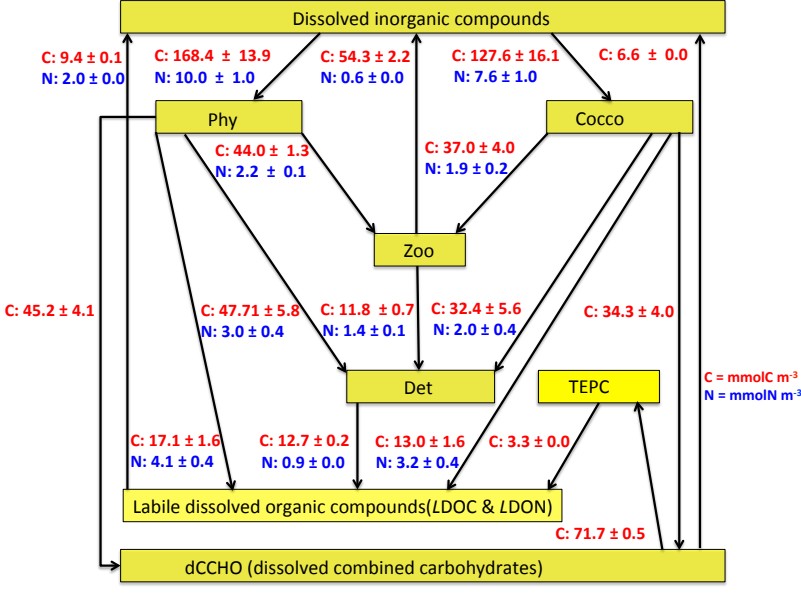

**(a) High CO$_2$**

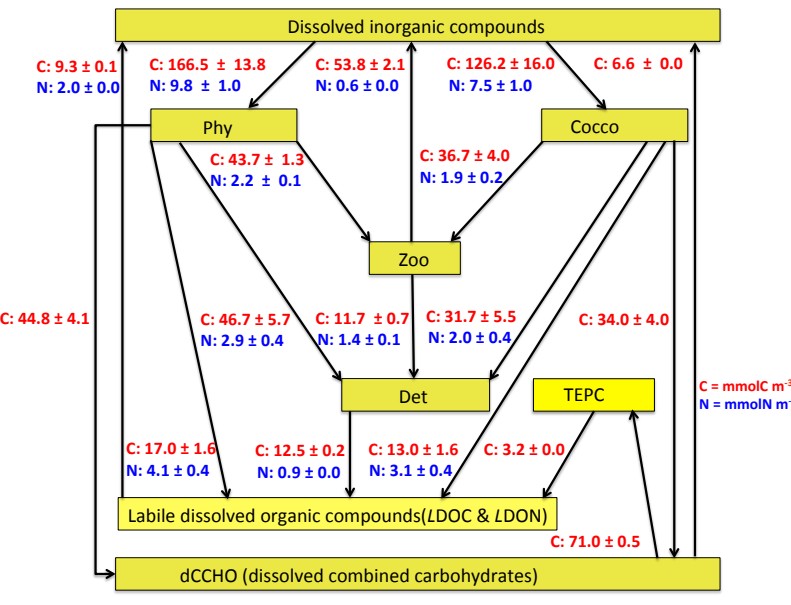

**(b) Low CO$_2$**

**Figure 12.** Carbon and nitrogen fluxes estimated by the model in mesocosms with high observed calcification but different CO$_2$ treatment, high (a) and low (b). All the arrows that point downwards show flux estimates from the respective compartment on the right hand side, whereas arrows pointing upwards show values on the left hand side.



**Figure 13.** Ratios of [POC]:[PON] and [DIC]:[DIN] determined from daily sampled noon values of model results. Filled circles represent $\log_{10}$ (DIC:DIN) ratios. Asteris symbols represent POC:PON ratio over the duration of the experiment.





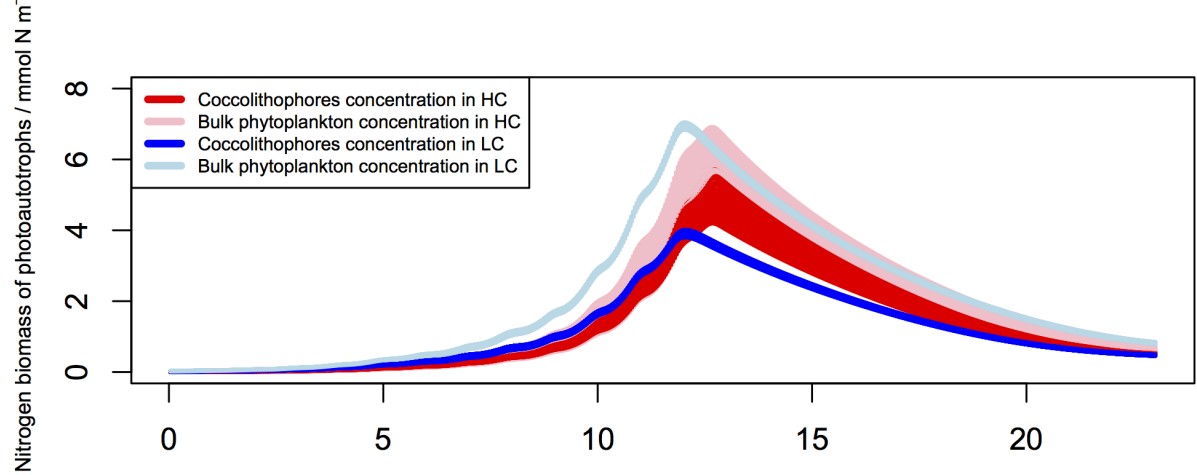

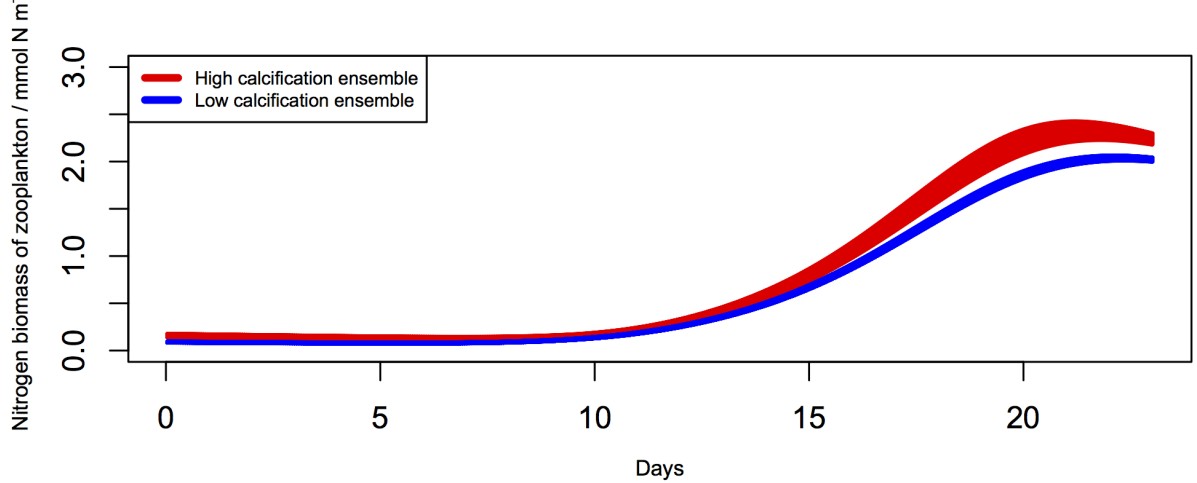

**Figure 14.** Simulated nitrogen biomass concentrations of photoautotrophs and zooplankton in high and low calcification solutions.





**Figure 15.** Simulated nitrogen biomass concentrations of coccolithophores in high, medium and low calcification solutions.





**Figure 16.** Bar plots depicting cumulative sum of PIC residual (modal-data misfit) from day 13 to day 18 of the experiment for three replicates in mean solution of HC, MC and LC ensembles. First row shows mesocosms with high $CO_2$ treatment (future), second row medium $CO_2$ treatment (present) and third row low $CO_2$ treatment (glacial).



## Observed and simulated variations in particulate inorganic carbon (PIC)

**Figure 17.** Full spread of model solutions according to credible range in parameter estimates, inlcuding ensemble solutions of high, medium and low calcification (light brown shaded area). Symbols represent observations of all mesocosms. Khaki shaded bands show $CO_2$ effect in the model, for solutions with lowest, medium and highest calcification rates.



## 7 Tables

| Auxiliary variables & functions | Description | Unit |
|---|---|---|
| $\mu_{cocco/phy}$ | net growth rates of respective photoautotrophs | $d^{-1}$ |
| $V^C$ | photoautotophic gross carbon fixation rates | mol C (mol N)$^{-1}$ d$^{-1}$ |
| $r^C$ | respiration rates | $d^{-1}$ |
| $V^N$ | carbon-specific nitrogen uptake rate | mol N (mol C)$^{-1}$ d$^{-1}$ |
| $Q^N$ | molar cellular nitrogen-to-carbon (N:C) ratio (cell quota) | mol N (mol C)$^{-1}$ |
| $S_I$ | degree of light saturation for photosynthesis | - |
| $f_V$ | resource fraction allocated for nutrient acquisition | - |
| $f_V^0$ | optimal allocation value of $f_V$ | - |
| $f_{LHC}^0$ | optimal resource allocation to light harvesting complex (LHC) | - |
| $f_{PIC}$ | calcification relative to net carbon fixation | mol PIC (mol C)$^{-1}$ |
| $f_{CO2}$ | regression model of $CO_2$ effect on calcification | - |
| $f_{dCCHO}$ | fraction of exudates assigned to dCCHO | - |
| $\theta$ | chlorophyll $a$-to-carbon (Chl:C) ratio of photoautotrophs | g Chl (mol C)$^{-1}$ |
| $\dot{\theta}$ | time derivative of $\theta$ | g Chl (mol C)$^{-1}$ d$^{-1}$ |
| $\theta^N$ | chlorophyll $a$-to-nitrogen (Chl:N) ratio of photoautotrophs | g Chl (mol N)$^{-1}$ |
| $A$ | nitrogen-specific rates of aggregation | mmol N m$^{-3}$ d$^{-1}$ |
| $G$ | nitrogen-specific rates of zooplankton grazing | mmol N m$^{-3}$ d$^{-1}$ |
| $r_{zoo}$ | zooplankton respiration | mmol C m$^{-3}$ d$^{-1}$ |
| $\gamma_{zoo}^N$ | zooplankton excretion of nitrogen | mmol N m$^{-3}$ d$^{-1}$ |
| $M_{zoo}$ | nitrogen-specific zooplankton mortality | mmol N m$^{-3}$ d$^{-1}$ |
| $T_f$ | Arrhenius temperature dependency | - |
| $F_{DIC}$ | flux due to air-sea gas exchange | mmol C m$^{-3}$ d$^{-1}$ |

| Model parameters (fixed) | | Value | |
|---|---|---|---|
| 1) $\gamma_N$ | photoautotrophic loss rate of organic nitrogen | 0.1 | $d^{-1}$ |
| 2) $C_{fact}$ | enhancement factor of carbon exudation relative to $\gamma_N$ | 1.0 | - |
| 3) $\rho$ | remineralisation rate of dissolved organic matter | 0.05 | $d^{-1}$ |
| 4) $\omega_{det}$ | hydrolysis/degradation rate of detritus | 0.02 | $d^{-1}$ |
| 5) $\omega_{gel}$ | hydrolysis/degradation rate of TEPC | 0.01 | $d^{-1}$ |
| 6) $\tau_{dissol}$ | dissolution rate of particulate inorganic carbon | 0.01 | $d^{-1}$ |
| 7) $\phi_{dCCHO}$ | coagulation parameter of dCCHO | $7.48 \cdot 10^{-4}$ | m$^3$ (mmol C)$^{-1}$ d$^{-1}$ |
| 8) $\phi_{TEPC}$ | coagulation parameter of dCCHO-TEPC | $2.56 \cdot 10^{-2}$ | m$^3$ (mmol C)$^{-1}$ d$^{-1}$ |

**Table 1.** Auxiliary model variables and model parameters. A complementary list of auxiliary variables and model parameters (that are not resolved in Eqs. 2 - 17) is given in Table (A.1) and detailed model equations are given in the Appendix (A).



| Initial conditions & parameters for optimisation | Description | Unit |
|---|---|---|
| 1) $PON_0$ | Initial concentration of particulate organic nitrogen | mmol N m$^{-3}$ |
| 2) $f_{det}$ | fraction of $PON_0$ assigned to non-living detritus | - |
| 3) $f_{zoo}$ | fraction of living $PON_0$ assigned to zooplankton | - |
| 4) $f_{cocco}$ | Initial coccolithophore fraction of photoautotrophs | - |
| 5) $Q_0$ | subsistence quota (minimum cellular N:C ratio) | mol mol$^{-1}$ |
| 6) $\alpha_{cocco}$ | Photosynthetic efficiency of coccolithophores | mol C (g Chl$a$)$^{-1}$ m$^2$ W$^{-1}$ d$^{-1}$ |
| 7) $\alpha_{phy}$ | Photosynthetic efficiency of non-calcifying phytoplankton | mol C (g Chl$a$)$^{-1}$ m$^2$ W$^{-1}$ d$^{-1}$ |

**Table 2.** Initial conditions and model parameters that are subject to optimisation.

| Parameter | Description | LC | MC | HC | Units |
|---|---|---|---|---|---|
| $PON_0$ | Parameter of initial PON concentration | 1.25 | 1.90 | 1.61 | mmol N m$^{-3}$ |
| $f_{det}$ | Parameter of initial detritus fraction | 0.89 | 0.89 | 0.89 | - |
| $f_{zoo}$ | Parameter of initial zoopl. fraction | 0.72 | 0.63 | 0.88 | - |
| $f_{cocco}$ | Parameter of initial coccolithophore fraction | 0.39 | 0.88 | 0.40 | - |
| $Q_0$ | Subsistence N:C ratio | $5.5 \cdot 10^{-2}$ | $4.2 \cdot 10^{-2}$ | $4.2 \cdot 10^{-2}$ | - |
| $\alpha_{cocco}$ | Photosynth. light absorpt. coeff. of coccolithoph. | 1.40 | 0.50 | 1.66 | mol C (g Chl $a$)$^{-1}$ m$^2$ W$^{-1}$ d$^{-1}$ |
| $\alpha_{phy}$ | Photosynth. light absorpt. coeff. of non-calcifiers | 1.73 | 3.10 | 1.71 | mol C (g Chl $a$)$^{-1}$ m$^2$ W$^{-1}$ d$^{-1}$ |

**Table 3.** Maximum likelihood parameter estimates of three model solutions: low, medium, and high calcification (LC, MC, and HC)



| | $f_{det}$ | $f_{zoo}$ | $f_{cocco}$ | $Q_0$ | $\alpha_{cocco}$ | $\alpha_{phy}$ |
|---|---|---|---|---|---|---|
| $PON_0$ | -0.03 / 0.03 / -0.30 | 0.57 / 0.48 / 0.51 | -0.10 / 0.29 / **0.66** | 0.05 / -0.20 / -0.34 | 0.11 / 0.03 / -0.56 | -0.10 / 0.19 / **0.60** |
| $f_{det}$ | 1 | -0.51 / -0.33 / **-0.92** | 0.13 / 0.01 / -0.28 | 0.23 / 0.25 / 0.11 | -0.15 / -0.10 / 0.10 | 0.13 / 0.03 / -0.40 |
| $f_{zoo}$ | . | 1 | -0.47 / 0.24 / 0.5 | -0.11 / -0.30 / -0.16 | 0.50 / 0.52 / -0.38 | -0.42 / 0.22 / **0.63** |
| $f_{cocco}$ | . | . | 1 | 0.10 / -0.12 / -0.25 | **-0.99** / -0.15 / **-0.95** | **0.99 / 0.93 / 0.93** |
| $Q_0$ | . | . | . | 1 | -0.10 / -0.25 / 0.18 | 0.13 / 0.10 / -0.26 |
| $\alpha_{cocco}$ | . | . | . | . | 1 | **-0.97** / -0.18 / **-0.87** |
| $\alpha_{phy}$ | . | . | . | . | . | 1 |

**Table 4.** Correlation coefficients of parameter estimates of low, medium, and high calcification model solutions (LC, MC, and HC). Correlation coefficients $\geq 0.6$ are marked bold face.

| State variable name | LC / mmol N m$^{-3}$ | MC | HC |
|---|---|---|---|
| $PON_0$ | $1.2 \pm 0.01$ | $1.9 \pm 0.01$ | $1.7 \pm 0.1$ |
| DetN$_0$ | $1.1 \pm 4 \cdot 10^{-4}$ | $1.7 \pm 1 \cdot 10^{-3}$ | $1.6 \pm 0.01$ |
| ZooN$_0$ | $0.1 \pm 1 \cdot 10^{-3}$ | $0.1 \pm 1 \cdot 10^{-3}$ | $0.2 \pm 0.01$ |
| CoccoN$_0$ | $0.02 \pm 2 \cdot 10^{-3}$ | $0.06 \pm 1 \cdot 10^{-3}$ | $0.01 \pm 2 \cdot 10^{-3}$ |
| PhyN$_0$ | $0.02 \pm 2 \cdot 10^{-3}$ | $0.01 \pm 4 \cdot 10^{-4}$ | $0.01 \pm 3 \cdot 10^{-3}$ |

**Table 5.** Mean initial values of PON ($PON_0$), detritus (DetN$_0$), zooplankton (ZooN$_0$), coccolithophores (CoccoN$_0$) and bulk phytoplankton (PhyN$_0$) according to posterior of the (initial condition) parameter estimates of three solutions: low, medium, and high calcification (LC, MC, and HC).



# Appendices

## A  Supplementary model equations

### A.1  Arrhenius relation

The affect of temperature on the metabolic rates and biological activities of the vast majority of organisms is given by the
Arrhenius relationship (Sibly et al., 2012).

$$T_f = \exp[-A_E \cdot (\frac{1}{T} - \frac{1}{T_{\mathrm{ref}}})] \tag{A.1}$$

where $T_{\mathrm{ref}}$ is reference temperature, given in units Kelvin (K). and approximately equals to 293.15 K (Table A.1).

### A.2  DIN uptake and carbon fixation

Under balanced optimal conditions we can approximate $f_V \approx f_V^0$ for photoautotrophs. The optimal allocation factor for LHC
in an algal cell is calculated from $f_V^0$ and $Q_0$:

$$Q_s = \frac{Q_0}{2} \tag{A.2}$$

$$f^0_{V_{phy/cocco}} = \frac{Q_s}{Q^N_{cocco/phy}} - \zeta^{\mathrm{N}} \cdot (Q^N_{cocco/phy} - Q_0) \tag{A.3}$$

$$f^0_{\mathrm{LHC}_{cocco/phy}} = 1 - \frac{Q_s}{Q^N_{cocco/phy}} - f^0_{V_{cocco/phy}} \tag{A.4}$$

where $\zeta^N$ is the cost of N uptake in a photoautotrophic cell, given in units mol mol$^{-1}$; and $Q_s$ is the N quota attached with
structural proteins, given in units mol N (mol C)$^{-1}$. In our model we do not make any differentiation between maximum N
assimilation rate and maximum carbon fixation rate. Therefore, both the quantities are identical.

$$V^N_{max} = V^C_{max} = V^N_0 \cdot T_f \tag{A.5}$$

where $V^N_{max}$ and $V^C_{max}$ are maximum N assimilation and maximum carbon fixation rates, given in units mol C (mol N)$^{-1}$.
Model parameter $V^N_0$ is photoautotrophic potential N fixation rate, given in units mol C (mol N)$^{-1}$ (Table A.1). The total N
uptake rate of photoautotrophs is calculated from the local N uptake rate (Pahlow et al., 2013). The latter is calculated from
maximum N assimilation rate, potential nutrient affinity and dissolved inorganic nitrogen concentration (DIN).

$$\hat{V}^N_{phy/cocco} = (\sqrt{\frac{1}{V^N_{max}}} + \sqrt{\frac{1}{A_0 \cdot (\mathrm{DIN})}})^{-2} \tag{A.6}$$





$$V_{phy/cocco}^N = f_{V_{cocco/phy}}^0 \cdot \hat{V}_{cocco/phy}^N \tag{A.7}$$

where $\hat{V}_{cocco/phy}^N$ is the local N uptake of photoautotrophs, given in units mol N (mol C)$^{-1}$. $A_0$ is potential nutrient affinity of respective algae in units m$^3$ (mol C)$^{-1}$ d$^{-1}$ (Table A.1).

The gross carbon fixation rate of calcifiers and non-calcifiers is calculated from day length, degree of light saturation, $f_{\text{LHC}}^0$ and $V_{max}^C$:

$$V_{cocco/phy}^C = L_d \cdot f0_{cocco/phy}^{\text{LHC}} \cdot V_{max}^C \cdot S_I^{cocco/phy} \tag{A.8}$$

where $V_{cocco/phy}^C$ is the gross carbon-fixation by photoautotrophs, given in units mol C (mol N)$^{-1}$, $L_d$ is the day length as
a fraction of 24 hours. For more details see Table (A.1). $S_I^{phy/cocco}$ is the degree of light saturation in photoautotrophs and calculated as:

$$S_I^{cocco/phy} = 1 - \exp(-\frac{\alpha \cdot \hat{\theta}_{cocco/phy} \cdot I}{V_0^C}) \tag{A.9}$$

$\hat{\theta}_{cocco/phy}$ is Chl:C ratio in the chloroplast of a cell (Pahlow and Oschlies, 2009; Pahlow et al., 2013).

The regulation term for chlorophyll $a$ synthesis ($S_{chl}$) is given as:

$$S_{chl} = \frac{\dot{\theta}_{cocco/phy}}{\theta_{cocco/phy}} = (\frac{1}{\zeta^{Chl}} \cdot \frac{\partial A_{cocco/phy}}{\partial \hat{\theta}_{cocco/phy}}) + (\dot{Q}_{cocco/phy}^N \cdot \frac{\hat{\theta}_{cocco/phy}}{\theta_{cocco/phy}} \cdot \frac{2 \cdot Q_s}{Q_{cocco/phy}^N \cdot Q_{cocco/phy}^N}) + \zeta^N \tag{A.10}$$

$$\frac{\partial A_{cocco/phy}}{\partial \hat{\theta}_{cocco/phy}} = L_d \cdot V_0^C \cdot [\frac{\alpha_{cocco/phy} \cdot I}{V_0^C} \cdot (1 - S_I^{cocco/phy}) \cdot (1 - \zeta^{Chl} \cdot \hat{\theta}_{cocco/phy}) - S_I^{cocco/phy} \cdot \zeta^{Chl}] - R_M^{Chl} \cdot \zeta^{Chl} \tag{A.11}$$

where, A is an auxiliary variable that contains all light dependent terms (Pahlow and Oschlies, 2009; Pahlow et al., 2013) and
has the unit d$^{-1}$; $\zeta^{Chl}$ and $\zeta^N$ are costs of chlorophyll $a$ synthesis and N assimilation, given in units mol C (g Chl)$^{-1}$ and mol C (mol N)$^{-1}$ (Table A.1). The derivative term ($\frac{\partial A}{\partial \hat{\theta}}$) is given in units mol C (g Chl)$^{-1}$ d$^{-1}$.

### A.3 Respiration costs

Total respiration cost in a cell includes costs due to chlorophyll synthesis, nutrient acquisition and cell maintenance.

$$r_{phy/cocco}^c = R_{phy/cocco}^{Chl} + \zeta^N \cdot V_{phy/cocco}^N + R_M \tag{A.12}$$

where respiration cost due to synthesis of chlorophyll $a$ is given as:

$$R_{phy/cocco}^{Chl} = (V_{phy}^C + f0_{phy/cocco}^{\text{LHC}} \cdot R_M^{Chl}) \cdot \zeta^{Chl} \cdot \hat{\theta}_{phy/cocco} \tag{A.13}$$

where $R_M$ is maintenance respiration cost of a cell, given in units d$^{-1}$. Detailed description of auxiliary variables is given in Tables (1 and A.1).



### A.4 Grazing

Equations below represent Holling type III grazing dynamics.

$$G_{phy} = g_m \cdot \frac{(\text{PhyN}^2)}{\epsilon + (\text{PhyN}^2)} \cdot \text{ZooN} \tag{A.14}$$

$$G_{cocco} = g_m \cdot \frac{(\text{CoccoN}^2)}{\epsilon + (\text{CoccoN}^2)} \cdot \text{ZooN} \tag{A.15}$$

where $g_m$ is the nitrogen specific maximum grazing rate on photoautotrophs, given in units $d^{-1}$; and $\epsilon$ is the half saturation constant for grazing, given in units $(\text{mmol N})^2 \, \text{m}^{-6}$.

### A.5 Zooplankton respiration and excretion

Respiration is parameterized as a function of respiration maintenance rate coefficient, temperature dependent metabolic rates and carbon concentration of heterotroph.

$$r_{zoo} = R_{basal} \cdot T_f \cdot \text{ZooC} \tag{A.16}$$

Similarly, excretion is parameterised as a function of respiration maintenance rate to basal metabolism, temperature dependent metabolic rates and nitrogen concentration of heterotroph.

$$\gamma_{zoo} = R_{basal} \cdot T_f \cdot \text{ZooN} \tag{A.17}$$

### A.6 Aggregation of photoautotrophic cells

Aggregation equations for bulk phytoplankton and coccolithophores are given below.

$$A_{phy} = \phi_{agg} \cdot \text{PhyN} \cdot \text{DetN} + \phi_{agg} \cdot \text{PhyN}^2 \tag{A.18}$$

$$A_{cocco} = \phi_{agg} \cdot \text{CoccoN} \cdot \text{DetN} + \phi_{agg} \cdot \text{CoccoN}^2 \tag{A.19}$$

### A.7 Regulation of calcification

Parameterisation of calcite-to-$C_{\text{organic}}$ ratio is given by Eq. (A.20), whereas regression model of Findlay et al. (2011) to quantify effect of different $CO_2$ concentrations on PIC formation is represented by Eq. (A.21).

$$f_{\text{PIC}} = \frac{1}{2} + \frac{s_{\text{PIC}}}{1 + \exp(s_{\text{PIC}} \cdot f0_{cocco}^{\text{LHC}})} \tag{A.20}$$

$$f_{\text{CO}_2} = -0.0097 \cdot \text{CO}_{2\,aq} + 0.9654 \tag{A.21}$$

with aqueous carbon dioxide $\text{CO}_{2\,aq}$ concentrations normalised to water mass instead of volume, given in units $\mu\text{mol kg}^{-1}$.





## A.8 dCCHO and TEPC

Given below is the parameterisation to estimate the fraction of phytoplankton exudates that become available to be part of dCCHO during two distinct modes of carbon overconsumption decscribed in Schartau et al. (2007).

$$f_{\text{dCCHO}}^{cocco/phy} = \left[ 1 + p_{\text{dCCHO}} \cdot \exp(1 - Q_s/Q_{cocco/phy}^N) \right]^{-1} \tag{A.22}$$

5  where $p_{\text{dCCHO}}$ is the fraction of DOC that enters dCCHO pool.

Coagulation parameter of dCCHO ($\phi_{\text{dCCHO}}$) is derieved from product of $\alpha_{\text{dCCHO}}$ (stickiness between dCCHO and dCCHO) and $\beta_{\text{dCCHO}}$ (C-specific collision rates between dCCHO and dCCHO). Likewise, coagulation parameter of dCCHO-TEPC ($\phi_{\text{TEPC}}$) is computed from the product of $\alpha_{\text{TEPC}}$ (stickiness between dCCHO and TEPC) and $\beta_{\text{TEPC}}$ (C-specific collision rates between dCCHO and TEPC). $\alpha_{\text{dCCHO}}$ and $\alpha_{\text{TEPC}}$ have no units as they are probabilities, whereas $\beta_{\text{dCCHO}}$ and $\beta_{\text{TEPC}}$ are given

10  in units $m^3$ (mmol C)$^{-1}$ d$^{-1}$. Values of $\alpha_{\text{dCCHO}}$, $\alpha_{\text{TEPC}}$, $\beta_{\text{dCCHO}}$ and $\beta_{\text{TEPC}}$ are taken from (Schartau et al., 2007).

$$\phi_{\text{dCCHO}} = \alpha_{\text{dCCHO}} \cdot \beta_{\text{dCCHO}}$$
$$\phi_{\text{dCCHO}} = (0.87 \cdot 10^{-3}) \cdot 0.86 = 7.48 \cdot 10^{-4} \tag{A.23}$$

$$\phi_{\text{TEPC}} = \alpha_{\text{TEPC}} \cdot \beta_{\text{TEPC}}$$
$$\phi_{\text{TEPC}} = 0.4 \cdot 0.064 = 2.56 \cdot 10^{-2} \tag{A.24}$$





| Auxiliary variables & functions | Description | Unit |
|---|---|---|
| $Q_s$ | N quota attached with structural proteins | mol N (mol C)$^{-1}$ |
| $V_{max}^N$ | photoautotrophic maximum N assimilation rates | mol C (mol N)$^{-1}$ d$^{-1}$ |
| $V_{max}^C$ | photoautotrophic maximum C fixation rates | mol C (mol N)$^{-1}$ d$^{-1}$ |
| $\hat{V}^N$ | photoautotrophic local N uptake rate of rate | mol N (mol C)$^{-1}$ d$^{-1}$ |
| $S_I$ | degree of light saturation for photosynthesis in photoautotrophs | - |
| $S_{chl}$ | regulation term for chlorophyll synthesis | mol C (mol N)$^{-1}$ |
| $L_d$ | day length as a fraction of 24 hours | - |
| $I$ | Mean irradiance | Wm$^{-2}$ d$^{-1}$ |
| $\hat{\theta}$ | photoautotrophic chloroplast Chl:C ratio | g Chl (mol C)$^{-1}$ |
| A | variable representing all light-dependent terms | d$^{-1}$ |
| $\alpha_{dCCHO}$ | stickiness between dCCHO and dCCHO | - |
| $\beta_{dCCHO}$ | C-specific collision rates between dCCHO and dCCHO | m$^3$ (mmol C)$^{-1}$ d$^{-1}$ |
| $\alpha_{TEPC}$ | stickiness between dCCHO and TEPC | - |
| $\beta_{TEPC}$ | C-specific collision rates between dCCHO and TEPC | m$^3$ (mmol C)$^{-1}$ d$^{-1}$ |

**Model parameters (fixed),**

| 1) to 8) are already listed in Table (1) | | Value | |
|---|---|---|---|
| 9) $T_{ref}$ | reference temperature for $A_E$ relation | 293.15 | K |
| 10) $A_E$ | slope of arrhenius relationship | 4500 | K |
| 11) $a_w$ | light attenuation due to water column | 0.04 | m$^{-1}$ |
| 12) $a_c$ | light attenuation due to chlorophyll $a$ | 0.05 | (mg Chl$a$)$^{-1}$ m$^3$ |
| 13) $R_M^{Chl}$ | cost of chlorophyll maintenance | 0.1 | d$^{-1}$ |
| 14) $R_M$ | total respiration maintenance cost | 0.05 | d$^{-1}$ |
| 15) $\zeta^{Chl}$ | cost of photosynthesis coefficient | 0.6 | mol C (g Chl$a$)$^{-1}$ |
| 16) $\zeta^N$ | cost of N uptake | 0.7 | mol C (mol N)$^{-1}$ |
| 17) $A_0$ | potential nutrient affinity | 1 | m$^3$ mol C$^{-1}$ d$^{-1}$ |
| 18) $V_0^N$ | photoautotrophic potential N fixation rate | 4.0 | mol C (mol N)$^{-1}$ |
| 19) $\gamma_N$ | algal nitrogen loss rate | 0.1 | d$^{-1}$ |
| 20) $\phi_{agg}$ | aggregation rate | 0.01 | m$^3$ (mmol N)$^{-1}$ d$^{-1}$ |
| 21) $p_{dCCHO}$ | minimum DOC fraction allocated to dCCHO | 0.2 | |
| 22) $g_m$ | nitrogen specific maximum grazing rate | 0.2 | d$^{-1}$ |
| 23) $\epsilon$ | prey capture rate normalised to maximum grazing rate | 1 | (mmol N)$^2$ m$^{-6}$ |
| 24) $M_{zoo}$ | mortality rate of zooplankton | 0.05 | d$^{-1}$ |
| 25) $R_{basal}$ | zooplankton basal respiration rate | 0.05 | d$^{-1}$ |
| 26) $p_{PIC}$ | slope of $\Delta$PIC formed per $\Delta$C assimilated | 5.0 | mol PIC (mol C)$^{-1}$ |

**Table A.1.** Complementary list of auxiliary variables and model parameters.



## B    Data correlation matrices

Correlations during pre-bloom ($t_i$; i = 1, ... , 13) between mesocosms with medium observed calcification in matrix form are given below:

$$
\boldsymbol{C}_{(\mathbf{y})} =
\begin{pmatrix}
 & \text{DIC} & \text{DIN} & \text{Chl}a & \text{PON} & \text{POC} & \text{PIC} & \text{TA} \\
\text{DIC} & 1 & 0.57 & -0.95 & -0.77 & -0.95 & -0.89 & 0.88 \\
\text{DIN} & . & 1 & -0.56 & -0.52 & -0.53 & -0.58 & 0.53 \\
\text{Chl}a & . & . & 1 & 0.71 & 0.91 & 0.81 & -0.77 \\
\text{PON} & . & . & . & 1 & 0.87 & 0.77 & -0.65 \\
\text{POC} & . & . & . & . & 1 & 0.83 & -0.77 \\
\text{PIC} & . & . & . & . & . & 1 & -0.95 \\
\text{TA} & . & . & . & . & . & . & 1
\end{pmatrix}
\tag{B.1}
$$

5    Correlations during post-bloom period ($t_i$; i = 14, ... , 22) are:

$$
\boldsymbol{C}_{(\mathbf{y})} =
\begin{pmatrix}
 & \text{DIC} & \text{DIN} & \text{Chl}a & \text{PON} & \text{POC} & \text{PIC} & \text{TA} \\
\text{DIC} & 1 & 0.22 & 0.27 & 0.29 & -0.83 & -0.93 & 0.94 \\
\text{DIN} & . & 1 & 0.3 & 0.31 & -0.23 & -0.22 & 0.24 \\
\text{Chl}a & . & . & 1 & 0.99 & 0.01 & -0.44 & 0.49 \\
\text{PON} & . & . & . & 1 & -0.02 & -0.45 & 0.50 \\
\text{POC} & . & . & . & . & 1 & 0.65 & -0.64 \\
\text{PIC} & . & . & . & . & . & 1 & -0.99 \\
\text{TA} & . & . & . & . & . & . & 1
\end{pmatrix}
\tag{B.2}
$$

Residual standard errors ($\sigma_i$) were calculated based on daily measurements between the mesocosms of similar observed calcification and can be written in matrix notation with off-diagonal elements being zero:

$$
\boldsymbol{S}_i =
\begin{pmatrix}
\sigma_i^{(\text{DIC})} & 0 & \cdots & 0 \\
0 & \sigma_i^{(\text{DIN})} & \cdots & \vdots \\
\vdots & \vdots & \ddots & 0 \\
0 & \cdots & 0 & \sigma_i^{(\text{TA})}
\end{pmatrix}
\tag{B.3}
$$




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
