# Peer review of "A data-model synthesis to explain variability in calcification observed during a CO2 perturbation mesocosm experiment"

_Biogeosciences, 2016_

## Referee Comment (RC1) · Anonymous Referee #1 · 28 Oct 2016

The manuscript submitted by Krishna and Schartau is targeted towards the use of mathematical modelling in order to understand the variability of biogeochemical data sets collected during 9 mesocosm experiments. The approach combined data analysis, model development and data assimilation and this combination of tools is a powerful way to gain a better understanding of an interdisciplinary data set. The idea is to gain a better understanding of the factors that may explain the variability in TA dynamics (and hence calcification) between the different mesocosms of a similar treatment. This is a really important topic since the impact of OA on calcification is still under debate. The current work shows that variability in TA could be explained by variations in initial conditions and the photo-acclimation states that prevailed within the phytoplankton

community during the filling of the mesocosm. The authors are rigorous in their approach and have the willingness to describe their model application extensively. This makes that in its present form, the manuscript is not really accessible to readers that are not expert in modeling. The methodology and the analysis of model results are described in a very detailed manner (see my comments below) and this sometimes prevents capturing the forest from the trees. The results section is, in some places, a succession of facts that are not enough integrated and may loss the reader. This is sometimes difficult to understand where the authors want to go and what is the additional information brought by the analysis of a particular fact (I had often to read by several time some paragraphs in order to capture the general idea). I would encourage the authors to strengthen the main messages of their study and their biogeochemical meaning. I would like them to clarify what is the added message compared to Eggers et al., (2014) who already stressed that variations in initial plankton composition can be responsible for large differences in the responses observed. Probably the new finding is that the photosynthesis efficiency of the community is (may) also important. However, as the authors correctly pointed out parameters may be collinear and this is not sure if a variation of less than 20 % of the photosynthesis efficiency as found by the authors (page 21, line 4) is really significant and does not compensate for a change in another parameter (to which alpha is co-linearly linked) that is not included in the 7 selected for the DA experiments. Besides, I would like to see the authors explain how their results are sensitive to the choice of the 7 parameters on which they decided to spend estimation effort. These parameters are selected without any clear justification (see my points below).

At the end we are expecting that the authors conclude on how their investigations bring an information on the potential impact of OA on calcification but this is missing.

Abstract: I suggest to improve the abstract. I find that lines 1-8 would be better placed in an introduction. Some parts of the remaining of the abstract is not really accessible to a non-specialized reader in DA.

Abstract line 13: "We explore how much of the observed variability in data can be explained by variations of initial conditions and by the effect of CO2 perturbations." I agree that this is exactly an important possible output of this type of study but unfortunately it is not enhanced enough in the manuscript. I would like to see a dedicated section/paragraph on that. (I suppose that by CO2 perturbation the authors are referring to OA?)

A table with the list of observations would be helpful and how it relates to the state variables

Page 4, line 30: .."The first is that we distinguish between bulk phytoplankton biomass and the presence of calcifying algae, coccolithophores like E. huxleyi"...It is not clear why this is an additional feature compared to what is mentioned before.

Page 12: line 9 : Âń we assume a higher C :N ratio (=2*6.625) only for initial detritus Âż, please add a justification.

Page 12: What are the "three distinct patterns in calcification"? I would not use attributable but observed. What do you mean by 'no such clear pattern"

Page 21: what do you mean by "adapting the same nomenclature" do you mean the same definition of partitioning of mescososms among different TA levels? It would be helpful to have some information on the general principles according to which this classification has been done. A significant part of the manuscript is based on the division of the mesocosm experiments in three main calcification levels and it would be appreciated that further justification is given as concern the statistical significance of the differences of the TA change between these three groups of mesocoms. (this has probably be done in other studies but some minimum justification would be appreciated).

Lines 13-16 would be better placed in the analysis of the mesocosm results and not in the design of the DA experiments. This paragraph is really not clear. Reading lines

16-20 does not help me to understand how data assimilation will be used in order to investigate the variability of TA. It seems that you will group the mesocoms according to their level of variation of TA and then? Please explain the general idea already here (I agree that it is somehow clarified afterwards).

In general, the section on DA needs to be reformulated. As it is now it is excessively complicated to understand why is data assimilation exactly used and what it will bring as a new information. The authors have to make that clear and to rewrite the technical description in order to target it to the audience of Biogeoscience which is not necessarily expert in technics like DA (you may also consider to put some materials in the appendix). It will also be very helpful to see further justifications for the choice of the 7 variables/parameters that are submitted to estimation (using DA). Considering the objective of the manuscript, this is surprising that parameters linked to calcification are not selected (e.g. fPIC, fPOC). Moreover, during the modeling experiments the authors realized that other parameters are important like C:Nfact , Chla:N but they are not added to the list.

Page 12, section 2.3.1: this section needs to be rewritten this is not understandable. line 26, observational residual errors, what is the cost function, R is not defined, ..

Why pCO2 and TEPC are not used in the observation vector?

Page 14, line 1: how do you estimate the daily residual standard errors?

Page 17, line 1: please give argument why this CN factor was not submitted to calibration since it seems that it is a very critical parameter. I find critical that as shown by Figure 8-10, model performances are not optimal for certain variables like chla, PIC, POC, PON, DIC after the bloom, it means exactly when we have variability of TA. This would require further justification by the authors for considering the model for assessing the TA dynamics during that period.

Page 17, line 23: How is estimated the standard error? Please specify (R terms ? )

Page 17, line 27: "First of all, from these flux estimates we learn that the CO2 effect introduced to the model, following Findlay et al. (2011), induces deviations in C flux that are much smaller than the variational range in model results, as reflected by the respective standard errors". This sentence is very difficult to understand. Please specify which CO2 effect you are referring? Which variation in C flux?

Page 18, line 33: "Carbon flux estimates show, carbon fixation in mesocosm with high CO2 treatment is slightly higher than in the mesocosm with low CO2 treatment". This difference is not significant if we consider the model error.

Page 19, line 4: "Our results show, regardless of biomass, coccolithophores are always less vulnerable to grazing than bulk phytoplankton". How does this fact result from model parameterization? The absence of data on zooplankton prevent a validation of this compartment and renders difficult the draw conclusion on the grazing.

Page 20, lines 21-24: "These considerations were disregarded when we designed this study and we originally thought of the importance of the relative mass distributions between the state variables resolved by our model, while imposing fixed initial stoichiometric ratios (C:N and Chla:N). It seems plausible to allow for some variations of the initial stoichiometric ratios as well." Do you mean that if you had to rebuild the model experiment you would change the list of 7 parameters?

Page 21, line 16: "Model biases and compensating effects are typically seen when applying DA methods (Bertino et al., 2003; Gregg, 2008)". This sentence is not clear. If it is necessary for the understanding of the rest of the paragraph, please clarify how DA can typically induce model bias and what are the compensating effects.

Figure 16: is the model able to differentiate the 3 groups of mesocosms (LC, MC, HC)? It seems that it overestimates calcification in the LC and underestimates it in the HC?

Minor comments

This is not clear why the salinity is decreasing during the course of the experiments. Is

it due to rainfall?

PAR is measured at the surface, which value of PAR is used to force the model

Page 5, line 21: the model maximizes

Page 5, line 27: "QN from", remove the "from"

Page 6: an algal

Page 6, line please define theta

Page 14, Line 12: better to use starting from published typical values rather than "while considering"

Page 16, line 24: please specify what do you mean by delta C? This is not defined (C-assimilation?)

Figure 5: please add TEPC and dCCHO.

Figure 7: please correct the ordinate: "calicification:c-assimilation" and not "gross carbon fixation-respiration".

---

## Referee Comment (RC2) · Anonymous Referee #2 · 21 Nov 2016

In this paper, a mathematical model is applied to a dataset consisting of 9 mesocosm experiments performed to elucidate the impact of CO2 enrichment on ecosystem development. It is shown that independent from the CO2 enrichment, the main distinction between the 9 mesocosms is in the degree in which there is calcification. Whereas this was already known from the literature, the model shows this to be mainly caused by the initial, physiological conditions of the coccoliths. These factors, not under experimental control, mask the possible impact of CO2 concentrations on ecosystem development, which was the original aim of the experimental study. In addition, the model results suggest that in the high CO2 scenario, there is impact of CO2 on the ecosystem to an extent that is not included by the model. As the model describes the effects of CO2

only on the relative amount of PIC produced, the deviance should be caused by other factors, yet unknown.

This is the second paper that uses a model on these data, but the emphasis of the current manuscript is different, the model used is also different, and has shown its worth in previous modelling studies.

Apart from the inevitable typo and small errors, I found the manuscript well written. The model equations are described at great length, and a rigorous data assimilation approach is taken, making the modelling results credible.

However, in its current state it is much too long which obscures the main messages. Due to the overkill of figures, tables, and text, it is also not simple make a good review – thus I decided to give mainly comments on how I would restructure the paper. Detailed comments can then be given on a second -shorter- version of the paper.

(1) I miss the ecological implications of the findings that have surfaced thanks to the model. For instance, the conclusions should not simply rephrase the modeling aspects, but discuss the ecological implications of the modeling study, or the consequences for future setups of such mesocosm studies.

(2) as a way to reduce the paper in size, I would remove all figures starting from Fig. 11 – and also significantly reduce (or remove) the corresponding text.

(3) I would also remove the lengthy CNfact discussion (and the corresponding figures 6 and 7, which are difficult to interpret).

In conclusion, I would suggest this manuscript to be acceptable but only after a strong reduction in size.

Some detailed comments.

P6 Line 24; symbol theta used but not explained.

Equation 13: I would have expected to see zooplankton excretion here.

p. 13-14 I do not understand how the three simulated mesocosms can be used to estimate data variances (Ri) for DIC, as this differs for each mesocosm?

P 15 L2-3. As far as I understand, the proposal distribution is adapted by the AMH algorithm Thus, the Hessian is used only as the initial proposal.

P1 L18; I do not understand what is meant with "optimal" mass flux estimates, as there are two completely different optimalisations: one for the data assimilation (DA), one for the optimal resource allocation in phytoplankton

Figures:

Fig. 2. Why is the Chl content not added here?

Fig. 3-4. Cumulative plots are difficult to interpret. I would prefer to see the actual probability distributions instead. Also, the figs 3 and 4 could be combined, if the three calcification scenarios for each parameter would be put in the same figure rather than in 3 of them.

Fig 8-10. The main difference is in between low and high calcification scenario – Fig 9. Could be removed.

Appendices: A7: the uptake rates are in mol N/mol C/day and not in mol N/molC. Similar for A8.

Table A1 – mean irradiance in W/m2/d -> should be W/m2

---

## Author Response (AR1)

**A data-model synthesis to explain variability in calcification observed during a $CO_2$ perturbation mesocosm experiment**

Shubham Krishna[1] and Markus Schartau[1]

[1]GEOMAR Helmholtz Centre for Ocean Research Kiel, Düsternbrooker Weg 20, 24105 Kiel, Germany

*Correspondence to:* Shubham Krishna (skrishna@geomar.de) and Markus Schartau (mschartau@geomar.de)

**Abstract.** A series of studies were conducted to investigate effects of ocean acidification (OA) on plankton dynamics. Among those were experiments with tanks or bags called mesocosms, with some enclosed water volume that typically comprised a natural plankton community. These mesocosms were typically perturbed and exposed to different carbon dioxide ($CO_2$) concentrations. Few studies focused on the impact of OA on growth of the coccolithophorid *Emiliania huxleyi*, a marine calcifying algae.

In our study we investigate data from a OA mesocosm experiment with *Emiliania huxleyi* and we apply an optimality-based model approach to study temporal changes and variability in observations, with focus on differences in total alkalinity (TA) and calcification. We explore how much of the observed variability in data can be explained by variations of initial conditions and by the effect of $CO_2$ perturbations. According to our model approach, changes in cellular calcite formation are resolved at the organism-level in response to variations in $CO_2$. With a data assimilation (DA) method we obtain estimates of initial conditions and of model parameters that determine photoautotrophic growth conditions. We compare ensembles of three distinctive model solutions that resolve low, medium and high calcification rates. Optimal estimates of the initial relative fraction of coccolithophores turned out to be correlated with estimates of the physiological model parameters. The spread of the optimised ensemble model solutions captures most of the observed variability. Optimised model solutions of the high $CO_2$ treatment are shown to systematically overestimate observed PIC production during a short period immediately after the maximum of the bloom. Hence, the $CO_2$ effect on calcification introduced to the model is insufficiently pronounced during this period. Our model results yield large differences in optimal mass flux estimates of carbon and of nitrogen even between mesocosms exposed to similar $CO_2$ conditions. Thus, our results show that small variations in initial abundance of coccolithophores and the prevailing physiological acclimation states between the individual mesocosms generate differences in calcification that are larger than the change in calcification induced by OA.

**Response to comments of Referee #1**

Krishna and Schartau 2016

*Correspondence to:* Shubham Krishna (skrishna@geomar.de) and Markus Schartau (mschartau@geomar.de)

5      We thank anonymous Referee #1 for detailed and useful comments. Many of the referee's suggestions will be considered. Below, we provide answers to all questions. With our responses we hope to sort out all ambiguities. Suggestions are made with respect to rewriting some paragraphs and restructuring the manuscript, e.g. by moving parts of the model description to the Appendix. Most of the proposed changes are currently applied to a revised manuscript version, mainly to ascertain feasibility.

10      **General comments by anonymous Referee #1:**

**Comment 1:** The authors are rigorous in their approach and have the willingness to describe their model application extensively. This makes that in its present form, the manuscript is not really accessible to readers that are not expert in modeling.

**Author's response:** We understand the referee's concern and we think that the model description can be shortened. However,
15   details of our model approach need to be accessible. We prefer to leave the general description of the processes resolved by the model in the main body of the text. The detailed dynamical model equations (mathematical description of all source minus sink terms) can be moved to the Appendix with little restructuring. This way we are able to shorten text in the subsection "Modelling approach" considerably. For example, the entire paragraph on optimal resource allocation (e.g. between light harvesting complex and sites of nutrient acquisition) may then appear in the Appendix.

**Comment 2:** The methodology and the analysis of model results are described in a very detailed manner (see my comments below) and this sometimes prevents capturing the forest from the trees. The results section is, in some places, a succession of facts that are not enough integrated and may loss the reader.

**Author's response:** We realised that it would not be critical to condense and remove content (text and figures) from the results
25   section. As proposed by Referee #2, we will merge Figs. (3) and (4). We admit that an explicit presentation of simulation results of transparent exopolymer particles (TEP) is not really relevant for our study and Fig. (6) can be removed. Furthermore, the entire discussion on possible differences between specific rate constants of carbon and nitrogen exudation is not needed to interpret the major outcome of the study. In fact, we learned that the consideration of a parameter $CN_{\text{fact}}$ that describes such possible difference is not fully consistent with the imposed assumptions of optimal resource allocation in our model. The value
30   of $CN_{\text{fact}}$ has to be equal to one, as assumed in our model approach in the first place. Thus, a parameter like $CN_{\text{fact}}$ is negligible in our case and it will be removed from our model and from this study.

Figure (9) (ensemble of medium calcification solutions) does not provide much additional information, given that Figs. (8) and (10) are already shown. We decided to remove Fig. (9), which was also recommended by Referee #2. Figure (15) was introduced to clarify that some model results are slightly biased, with a build-up of coccolithophore biomass that is too fast.
35   We eventually decided to not to omit Fig. (15) but refined it.

The removal of Figs. (6, 9), the combination of Figs. (3 and 4), makes the study more concise.

**Comment 3:** I would like them to clarify what is the added message compared to Eggers et al., (2014) who already stressed that variations in initial plankton composition can be responsible for large differences in the responses observed.

**Author's response:** We will consider this aspect and will revise the text. Briefly, our results not only support the findings of Eggers et al. (2014), they provide additional insight to the problem of resolving a $CO_2$ response in the presence of variability in measurements. With our analysis we make inference about the linkage between phytoplankton growth dynamics, calcification, variability in observations and uncertainties in model results. We determine the conditional probability of how well the experimental data can be represented by our dynamical model approach. The analysis of an ensemble of statistically equivalent model solutions (according to maximum likelihood estimates) differs from a statistical treatment and analysis of the experimental data, e.g. as described in Eggers et al. (2014). Our study includes a mechanistic description of algal growth, thereby resolving nonlinearities with respect to carbon and nitrogen acquisition and chlorophyll $a$ synthesis. Based on our data assimilation approach we obtain maximum likelihood estimates of important model parameters. These estimates determine model solutions of carbon and nitrogen flux for all nine mesocosms. The model resolves temporal variations of specific variables that are not resolved by the available measurements (e.g. diurnal variations).

One added message is that our mass flux estimates are shown to differ more between the different calcification solutions than between the different $CO_2$ treatments. This situation exemplifies that simulation results (e.g. future model projections) may involve uncertainties in flux estimates that are larger than the $CO_2$ effect introduced to the model (e.g. by following Findlay et al., 2011). Another added message is that initial conditions may not be independently estimated from estimates of phytoplankton growth parameters, like $\alpha_{phy}$ and $\alpha_{cocco}$. This is particularly relevant for model assessment and model analyses of mesocosm experiments.

Some other message is that the original design of the experiment was meaningful, in particular with respect to the initial filling of the mesocosms. The retrospective separation the $CO_2$ response signal from the system's variability was only possible because mesocosms with similar initial conditions were subject to different $CO_2$ concentrations. Such separation would be more difficult in retrospective if mesocosms with similar initial conditions would have been (by chance) exposed to similar $CO_2$ levels.

To facilitate data analysis of a mesocosm experiment it is meaningful to 1) analyse details of initial conditions (e.g. $f_{cocco}$ and $f_{zoo}$) prior to perturbation (assuring that similar replicate mesocosms do not get exposed to identical $CO_2$ levels), 2) perform side experiments that can reveal the photosynthetic efficiency during the exponential growth phase.

We will revise parts of the discussion and conclusion section in order to straighten the above points.

**Comment 4:** However, as the authors correctly pointed out parameters may be collinear and this is not sure if a variation of less than 20 % of the photosynthesis efficiency as found by the authors (page 21, line 4) is really significant and does not compensate for a change in another parameter (to which alpha is co-linearly linked) that is not included in the 7 selected for the DA experiments.

**Author's response:** A discussion on possible collinearities of $\alpha_{cocco}$ with some other fixed parameter is not helpful here. However, the referee has raised some important point that needs attention and should be better described in the text. The estimates of $\alpha_{cocco}$ are negatively correlated with the estimates of $f_{cocco}$ and with $\alpha_{phy}$ (see Table 4). With respect to the initial abundance of coccolithophores we may add few sentences for clarification. We suggest to depict relevant combinations of both parameters $\alpha_{cocco}$ and $f_{cocco}$ to explain the major differences between the low and high calcification solution. The revised text (highlighted in red) would then read as follows:

"Major differences between the LC and HC solutions can thus be attributed to higher $\alpha_{cocco}$ values (median $\alpha_{cocco}$ = 1.7 mol C (g Chl $a$)$^{-1}$ m$^2$ W$^{-1}$ d$^{-1}$) in HC posterior distribution compared to LC (median $\alpha_{cocco}$ = 1.4 mol C (g Chl $a$)$^{-1}$ m$^{-1}$ d$^{-1}$). The estimates of $\alpha_{cocco}$ are negatively correlated with the estimates of $f_{cocco}$ (Table 4) and we may therefore look on the combination of the two parameters. To do so we compare two extreme solutions, selected from the ensemble solutions of LC and HC respectively. One extreme solution yields the lowest calcification among all HC solutions, based on the parameter combination ($\alpha_{cocco}$ = 1.84 mol C (g Chl $a$)$^{-1}$ m$^2$ W$^{-1}$ d$^{-1}$ , $f_{cocco}$ = 0.34). The other selected solution represents the highest calcification of all LC solutions, which corresponds with ($\alpha_{cocco}$ = 1.59 mol C (g Chl $a$)$^{-1}$ m$^2$ W$^{-1}$ d$^{-1}$ , $f_{cocco}$ = 0.35). Thus, it is mainly the photosynthetic efficiency $\alpha_{cocco}$ to which the model solution is highly sensitive to. Hence, a difference of $\approx 0.3$ mol C (g Chl $a$)$^{-1}$ m$^2$ W$^{-1}$ d$^{-1}$ can effectively determine the differences in our simulations with respect to rates of carbon fixation and calcification. "

**Comment 5:** I would like to see the authors explain how their results are sensitive to the choice of the 7 parameters on which they decided to spend estimation effort. These parameters are selected without any clear justification.

**Author's response:** A full description of the parameter selection process would go beyond the scope of our study presented here. Furthermore, there is no purely objective reasoning, but the decision is based on results from preparatory analyses. Although we are obliged to reduce content of the method section, we think it is worthwhile adding few sentences in this respect: "The decision on which parameters should become subject to optimisation is based on a series of preceding parameter optimisations and subsequent sensitivity analyses. A major objective is to reduce the number of parameters for optimisation to a meaningful minimum. This facilitates the identification of those parameter values that are of primary concern. Since we address differences in initial conditions in our study, we consider four parameters that determine these differences and they need to become subject to optimisation. The additionally selected three growth parameters are amongst those to which the model solution is most sensitive. The model solutions are also highly sensitive to variations of the maximum potential nitrogen uptake rate ($V_0^m$). This parameter is excluded from optimisation, because it is not possible to obtain estimates of ($V_0^m$) that are independent of estimates of the photosynthetic efficiency. Therefore, a value is assigned to $V_0^m$ that is typical and was used for simulations of other experiments (e.g. Pahlow et al., 2013), ensuring credible estimates of those parameters that are

optimised in our study. The mesocosm experiment covers only a short post-bloom period and we found other parameters, like maximum grazing rates and the aggregation parameters, to be weakly constrained by the available data. Their consideration for optimisation would impede the identification of the other more important parameters. Values assigned to those parameters that are excluded from optimisation are adapted from other studies (e.g. Pahlow et al., 2013; Schartau et al., 2007). ”

**Comment 6:** At the end we are expecting that the authors conclude on how their investigations bring an information on the potential impact of OA on calcification but this is missing.

**Author's response:** We will introduce an extra paragraph to the conclusion section, further emphasising the implications of our results (see above response to Comment 3). We will also include some suggestions for setup of future mesosocosm studies
10   on ocean acidification.

The revised paragraph (shown in red) in the end of conclusion section reads:

" An analysis of data of a mesocosm experiment is often approached by first grouping individual mesocosms according to the level of perturbation (e.g. the level of DIC added). In some cases, such apparently self-evident approach may not help to reveal
15   some basic phenomenon in mesocosm experiments. For a meaningful data analysis the mesocosms need not be exclusively differentiated by the different levels of perturbation but may first be sorted by major differences between relevant response signals, as done with respect to the magnitude of calcification in our study (by differentiating between LC, MC, and HC). In mesocosm experiments these differences in responses are likely associated with variations in initial conditions.

With our DA approach we could disentangle three distinctive ensembles of model solutions that represent mesocosms with
20   high, medium and low calcification rates. The results of our data-model synthesis show that the initial relative abundance of coccolithophores and the prevailing physiological acclimation states drive the bloom development and determine the amount of calcification in the mesocosms. Small variations of these two initial factors between the mesocosms can generate differences in calcification that are larger than the change in calcification induced by OA. In spite of this difficulty, a $CO_2$ response signal may still be identifiable, as long as mesocosms that reveal strongest similarities (with respect to initial composition of plankton
25   and their physiological state) are not used as replicates for similar $CO_2$ conditions (perturbations). Instead, mesocosms with similar initial conditions should be exposed to different levels of OA. Such favourable starting conditions were met in the mesocosm experiment described in Engel et al. (2005) and Delille et al. (2005), as well as in the experiment of Eggers et al. (2014).

An alternative approach to setting up mesocosms is to gradually increase the level of perturbation for a series of mesocosms.
30   This way a gradient of different perturbation levels is introduced. The advantage then is that mesocosms that have been collated according to e.g. lowest and highest response signals (or likewise according to similarities in initial conditions) may then be separately analysed with respect to their responses to the individual levels of perturbation. ”

*Specific comments by anonymous Referee # 1:*

**Specific comment 1:** I suggest to improve the abstract. I find that lines 1-8 would be better placed in an introduction. Some parts of the remaining of the abstract is not really accessible to a non-specialized reader in DA.

**Author's response:** We will consider this suggestion.

**Specific comment 2:** Abstract line 13: "We explore how much of the observed variability in data can be explained by variations of initial conditions and by the effect of CO2 perturbations." I agree that this is exactly an important possible output of this type of study but unfortunately it is not enhanced enough in the manuscript. I would like to see a dedicated section/paragraph on that. (I suppose that by CO2 perturbation the authors are referring to OA?)

**Author's response:** Various aspects of "variability" are explicitly addressed in the manuscript: Sect. (3.2) Data-model comparison, Sect. (4.1) Uncertainty ranges in parameter estimates and variability in model solutions, Sect. (4.3) Disentangling $CO_2$ effect from the observed variability in PIC.

Ocean acidification (OA) is a wide and general term. In the sentence we refer specifically to a mesocosm $CO_2$ perturbation experiment. We do not find it appropriate tto change the term "$CO_2$ perturbation" to OA.

**Specific comment 3:** A table with the list of observations would be helpful and how it relates to the state variables.

**Author's response:** A list of observations and how it relates to the state variables is depicted on page 13, see Eq. (25). Abbreviations and units are described in the corresponding text. Note that Sect. (2.3.1) will be restructured, see responses to specific comments 11 and 15.

**Specific comment 4:** Page 4, line 30: ... "The first is that we distinguish between bulk phytoplankton biomass and the presence of calcifying algae, coccolithophores like E. huxleyi". It is not clear why this is an additional feature compared to what is mentioned before.

**Author's response:** Yes, we agree. This is redundant information and we will therefore delete the sentence.

**Specific comment 5:** Page 12, line 9: "We assume a higher C :N ratio (=2*6.625) only for initial detritus". Please add a justification.

**Author's response:** Since the mesocosms were filled with post-bloom, nutrient depleted water masses, we assume that all dead particulate organic matter has a C:N ratio that is rather typical for such post-bloom conditions.

**Specific comment 6:** What are the "three distinct patterns in calcification"? I would not use attributable but observed. What do you mean by "no such clear pattern".

**Author's response:** For clarification we will include a new figure (uploaded as supplement material of our response). The left panel in the figure shows three distinct calcification patterns, reflected in total alkalinity (TA) data. Those mesocosms that exhibit high TA values (a reduced drawdown during the bloom and post-bloom period) feature rates of low calcification (LC,

in blue color). Mesocosms with low TA values (a strong reduction of TA) reveal rates of high calcification (HC, marked red). Rates of medium calcification (MC) are assigned to the remaining mesocosms (with intermediate TA values, marked black). The right panel shows the respective different $CO_2$ treatments in the same colors as for LC, MC, and HC. The figure shows that each calcification case (LC, MC, and HC) includes mesocosm of all three $CO_2$ treatments.

5 **Specific comment 7:** Page 12, What do you mean by "adapting the same nomenclature", do you mean the same definition of partitioning of mescoms among different TA levels? It would be helpful to have some information on the general principles according to which this classification has been done.

**Author's response:** By "adapting the same nomenclature" we simply mean the names given to respective mesocosms are the identical with the names in Engel et al., 2005 and Delille et al., 2005. For example, mesocosm one is referred as M1, mesocosm
10 two as M2, so on and so forth.

**Specific comment 8:** Page 12, A significant part of the manuscript is based on the division of the mesocosm experiments in three main calcification levels and it would be appreciated that further justification is given as concern the statistical significance of the differences of the TA change between these three groups of mesocoms. (this has probably be done in other studies
15 but some minimum justification would be appreciated).

**Author's response:** Statistical significance of the differences in TA changes was not tested in Delille et al. (2005). Both studies (Delille et al., 2005, and Engel et al., 2005) rather focused on statistical significance between the $CO_2$ treatments. The mesocosms were pooled according to the $CO_2$ levels, thereby including all variability in calcification.

20    If the differences in TA between the LC, MC, and HC pooled mesocosms were insignificant we would see this in our maximum likelihood estimates of the model parameters as well. This is because we consider mean data, standard errors, and the correlations between the different observational types (of the pooled mesocosms) in the cost function.

During the post-bloom period, the mesocosms pooled in HC reveal TA changes that are consistently higher than in the LC
25 mesocosms. In fact, these differences become well reflected in our parameter estimates. Thus, our optimised ensemble model solutions are providing the statistical evidence that HC and LC are significantly different.

With respect to the mesocosms assigned to the MC (medium calcification) case we see in our parameter estimates and ensemble model solutions that they are rather close to conditions also met by the HC mesocosms. In this case the differences in
30 parameter estimates (between MC and HC) are small, although we find significantly different estimates for $\alpha_{cocco}$ and for $f_{zoo}$ between MC and HC (see Figs. 3 and 4). Thus, we may have one or two out of the three MC mesocosms that might have been better assigned to the HC case. However, this is reflected in our DA results and we are primary concerned with the upper and lower extremes in calcification, as resolved by the six mesocosms in the LC and HC cases.

**Specific comment 9:** Page 12, lines 13-16 would be better placed in the analysis of the mesocosm results and not in the design of the DA experiments. This paragraph is really not clear.

**Author's response:** It is not meaningful to move this to the results section. Since this seems to be the core of confusion we suggest introducing a figure that documents the differences between LC, MC, and HC mesocosms. We will revise the text on page 12 accordingly. The following points will be further clarified: 1) the selection of mesocosms assigned to LC, MC and HC is entirely based on the observational data, 2) we simulate all (three) mesocosms of each case LC, MC, HC, 3) in the cost function we compare the daily means of model results (of those mesocosms of the respective calcification case) with the observed means (of the same mesocosms), 4) we thus obtain parameter estimates for the cases LC, MC, and HC.

**Specific comment 10:** Page 12, Reading lines 16-20 does not help me to understand how data assimilation will be used in order to investigate the variability of TA. It seems that you will group the mesocoms according to their level of variation of TA and then? Please explain the general idea already here (I agree that it is somehow clarified afterwards).

**Author's response:** We think we have addressed this in our responses to the specific comments 6), 8) and 9).

**Specific comment 11:** In general, the section on DA needs to be reformulated. As it is now it is excessively complicated to understand why is data assimilation exactly used and what it will bring as a new information. The authors have to make that clear and to rewrite the technical description in order to target it to the audience of Biogeoscience, which is not necessarily expert in technics like DA (you may also consider to put some materials in the appendix).

**Author's response:** The subsection on data assimilation is an essential part of the study and it describes important aspects. We again critically reviewed the subsection and we find the description appropriate with respect to accessibility, style and content.

Equations (24, 25 and 26) provide important information, since definitions of the cost function are the major integral part have been different studies, which has consequences for parameter estimates. However, we will restructure parts of Sect. (2.3.1), see response to specific comment 15 below.

Some of the complexity of the actual optimisation procedure is reflected in the subsection "Parameter optimisation procedure". As this might only be of interest for those readers who are involved in applying similar approaches, we suggest moving this subsection (Sec. 2.3.2) to the Appendix entirely.

**Specific comment 12:** It will also be very helpful to see further justifications for the choice of the 7 variables/parameters that are submitted to estimation (using DA).

**Author's response:** This comment conforms to the general comment 5) above. You may refer to our answer above, the response provided to the respective comment.

**Specific comment 13:** Considering the objective of the manuscript, this is surprising that parameters linked to calcification are not selected (e.g. fPIC, fPOC).

**Author's response:** $f_{PIC}$ is an auxiliary variable (see Table 1) and not a parameter. To our model we have not introduced a

parameter referred to as $f_{POC}$.

**Specific comment 14:** Moreover, during the modeling experiments the authors realized that other parameters are important like CNfact , Chla:N but they are not added to the list.

5 **Author's response:** $CN_{fact}$ was included in Table (1). Some confusion might have been caused by a small typo that we found in the text: the parameter name is $CN_{fact}$ not $C_{fact}$. This parameter will be removed from the study and its discussion is not meaningful, as pointed out before in the response to general Comment 2.

$\theta^N$ (Chl$a$:N ratio) is an auxiliary variable (see Table 1). It expresses the photoacclimation state. It varies with time and we therefore do not need to estimate a constant value for it.

**Specific comment 15:** Page 12, section 2.3.1: this section needs to be rewritten this is not understandable. Line 26, observational residual errors, what is the cost function, R is not defined.

**Author's response:** We will revise parts of Sect. (2.3.1). Some restructuring should make it easier to understand. The obser-
15 vation vector $\boldsymbol{y}_i$ can be introduced first together with its model counterpart ($H_i(\boldsymbol{x})$). This will be followed by the equation that represents the residuals between data and model results (currently given as Eq. 25). We may then explain the derivation of the cost function $J(\Theta)$ (Eqs. 23 and 24). Thereafter, the calculation of the covariances ($R$) will be explained, with reference to details given in the Appendix.

20 **Specific comment 16:** Why pCO$_2$ and TEPC are not used in the observation vector?

**Author's response:** We have included dissolved inorganic carbon (DIC) in the observation vector together with total alkalinity, which accounts for similar information as pCO$_2$. Thus, no additional independent information would be introduced to the cost function if pCO$_2$ data were added.

With regard to TEP, we found few discrepancies between TEP data depicted in Joassin et al. (2011) and Engel et al. (2005),
25 and the measurements available from PANGAEA data library. We could not resolve this problem and have therefore decided not to assimilate TEPC (Alcian blue concentrations converted to carbon units) into our model but to compare the typical concentration range only.

**Specific comment 17:** Page 14, line 1: how do you estimate the daily residual standard errors?

30 **Author's response:** The term "residual" will be removed, as it can be confused with the residuals between the data and the model results. For every calcification case we calculated daily standard error: standard deviation ($\sigma_{std}$) divided by the square root of the number of samples (mesocosms) available on that particular day ($n$).

**Specific comment 18:** Page 17, line 1: Please give argument why this CN factor was not submitted to calibration since it seems
35 that it is a very critical parameter. I find critical that as shown by Figure 8-10, model performances are not optimal for certain

variables like chla, PIC, POC, PON, DIC after the bloom, it means exactly when we have variability of TA. This would require further justification by the authors for considering the model for assessing the TA dynamics during that period.

**Author's response:** The model parameter $CN_{fact}$ will be removed from the study and its discussion is not so meaningful, as described before in the response to general comment 2. Our model represents basic biogeochemical and eco-physiological processes related to plankton dynamics in a mesocosm experiment. We show in our results (Fig. 5) that the model does reproduce most of the data. Given the complex dynamics involved, the model performance is very good. Due to the simplifications introduced, any model will remain imperfect and will be limited in resolving the complexity of a plankton ecosystem. In Sect. (4.2) we have explicitly stressed systematic model deficiencies and discuss these biases.

**Specific comment 19:** Page 12, line 27: How is estimated the standard error? Please specify (R terms?)

**Author's response:** How standard errors are calculated is described in the response to Comment 17, above. The correlation matrices for the exponential growth phase and post-bloom period are given in the Appendix, see Eqs. (B1 and B2).

**Specific comment 20:** Page 17, line 27: "First of all, from these flux estimates we learn that the CO2 effect introduced to the model, following Findlay et al. (2011), induces deviations in C flux that are much smaller than the variational range in model results, as reflected by the respective standard errors". This sentence is very difficult to understand. Please specify which CO2 effect you are referring? Which variation in C flux?

**Author's response:** The regression model of Findlay et al., (2011) is implemented in our model to quantify the effect of different $CO_2$ perturbation on PIC formation, which we refer as $CO_2$ effect (as given in Eq. A21). We show in our study (Figs. 11 and 12) that simulated carbon and nitrogen mass flux estimates differ more between the mesocosms with different calcification rates than between the mesocosms with the different $CO_2$ treatments.. However, we do agree with the referee that the line, "as reflected by the respective standard errors ",might be misleading. Therefore, we will remove the same line..

**Specific comment 21:** Page 17, line 33: "Carbon flux estimates show, carbon fixation in mesocosm with high CO2 treatment is slightly higher than in the mesocosm with low CO2 treatment". This difference is not significant if we consider the model error.

We fully agree with the referee's comment. Therefore, we will remove the above quoted sentence from our manuscript.

**Specific comment 22:** Page 19, line 4: "Our results show, regardless of biomass, coccolithophores are always less vulnerable to grazing than bulk phytoplankton". How does this fact result from model parameterization? The absence of data on zooplankton prevent a validation of this compartment and renders difficult the draw conclusion on the grazing.

**Author's response:** We agree with the referee's comment. No solid conclusions can be drawn with respect to differences in grazing. This is because we have no information about the grazing rates. The sentence expresses model behaviour. To avoid any misunderstanding we suggest rephrasing the sentence:

" According to our model solutions, the coccolithophores are always less vulnerable to grazing than the bulk phytoplankton. This model behaviour may not be representative or conclusive, because we have no information about the actual grazing rates or about grazing preferences. "

**Specific comment 23:** Page 20, lines 21-24: "These considerations were disregarded when we designed this study and we originally thought of the importance of the relative mass distributions between the state variables resolved by our model, while imposing fixed initial stoichiometric ratios (C:N and Chla:N). It seems plausible to allow for some variations of the initial stoichiometric ratios as well". Do you mean that if you had to rebuild the model experiment you would change the list of 7 parameters?

**Author's response:** Correct, now that we have indications of the initial acclimation state being important, we would set up initial C:N and Chl$a$:N ratios as additional parameters for optimisation. Thus, the number of parameters would increase from 7 to 9.

**Specific comment 24:** Page 21, line 16: "Model biases and compensating effects are typically seen when applying DA methods (Bertino et al., 2003; Gregg, 2008)". This sentence is not clear. If it is necessary for the understanding of the rest of the paragraph, please clarify how DA can typically induce model bias and what are the compensating effects.

**Author's response:** The revised sentence reads:

"Model biases disclose systematic deviations of simulation results from observations, which may point towards i) erroneous model counterparts to observations (definition of $H(\boldsymbol{x})$ in Eq. 25) or ii) deficiencies in model dynamics (errors in $\boldsymbol{x}$). A noticeable bias is related to the increase in PON concentration during the late phase of exponential growth (between days 10 and 12). This offset... "

**Specific comment 25:** Figure 16: Is the model able to differentiate the 3 groups of mesocosms (LC, MC, HC)? It seems that it overestimates calcification in the LC and underestimates it in the HC?

**Author's response:** As described in response to specific comment 8, we see differences in the maximum likelihood estimates of the model parameters for the mesocosms of LC, MC and HC. Further, we show in Figs. (8) and (10) of our manuscript that the model successfully reproduces high and low observed PIC values. However, the referee rightly pointed out that the model overestimates calcification in LC case (especially for the mesocosm with high $CO_2$ treatment). This issue has been addressed in the Sect. (4.3) of our manuscript. In the HC case, the observed PIC values are well in range of high calcification model solutions (ensemble spread).

*Minor comments by anonymous Referee # 1:*

**Minor comment 1:** This is not clear why the salinity is decreasing during the course of the experiments. Is it due to rainfall?

**Author's response:** We have not cross-checked for why salinity decreases. We did not find any explanation in the respective publications, neither in Delille et al. (2005) nor in Engel et al. (2005).

All other minor comments and corrections suggested by anonymous Referee #1 will be implemented.
* * *
**Response to comments of Referee #2**

We thank Referee # 2 for the valuable feeback. We will consider most of referee's suggestions for revising our manuscript.
10  One suggestion is to significantly reduce the size of the manuscript. We understand the referee' s constructive criticism. It corresponds with a concern mentioned by the other referee. We largely agree and think that the manuscript can be shortened considerably. For example, all dynamical model equations (detailed source minus sink terms) can be moved to the Appendix. We also decided to put Sect. (2.3.2, Parameter optimisation procedure) to the Appendix, as this might appear to be too difficult to understand for readers who are not familiar with optimisation problems. Those results that are not directly related to
15  calcification or to the build-up of coccolithophore biomass will be sorted out.

We will also consider the referee's suggestion to remove the discussion on $CN_{\mathrm{fact}}$, because we have learned that it would actually require an extended in-depth discussion. This is because a $CN_{\mathrm{fact}}$ that is different from *one* does not exactly comply with the model's optimality assumptions applied in this study. We greatly appreciate the referee's critical evaluation of our discussion on $CN_{\mathrm{fact}}$. Some figures will be removed from the results and discussion sections. We will, as suggested, combine
20  Figs. (3 and 4), although we prefer plotting the cumulative probability distributions rather than showing the probability densities (see response to specific comment 8). Parts of the conclusion section will be rewritten.

**General comments by anonymous Referee #2:**

25  **Comment 1:** I miss the ecological implications of the findings that have surfaced thanks to the model. For instance, the conclusions should not simply rephrase the modeling aspects, but discuss the ecological implications of the modeling study, or the consequences for future setups of such mesocosm studies.

**Author's response:** We agree with Referee # 2. In our conclusion section we have not sufficiently stressed the ecological implication of our study and we may also bring forward some general suggestions with respect to the design of future mesocosm
30  "perturbation" studies. In a revised version we will have an extra paragraph added to the conclusion section. The additional paragraph (shown in red) could read:
"

An analysis of data of a mesocosm experiment is often approached by first grouping individual mesocosms according to the level of perturbation (e.g. the level of DIC added). In some cases, such apparently self-evident approach may not help to reveal
35  some basic phenomenon in mesocosm experiments. For a meaningful data analysis the mesocosms need not be exclusively

differentiated by the different levels of perturbation but may first be sorted by major differences between relevant response signals, as done with respect to the magnitude of calcification in our study (by differentiating between LC, MC, and HC). In mesocosm experiments these differences in responses are likely associated with variations in initial conditions.

With our DA approach we could disentangle three distinctive ensembles of model solutions that represent mesocosms with high, medium and low calcification rates. The results of our data-model synthesis show that the initial relative abundance of coccolithophores and the prevailing physiological acclimation states drive the bloom development and determine the amount of calcification in the mesocosms. Small variations of these two initial factors between the mesocosms can generate differences in calcification that are larger than the change in calcification induced by OA. In spite of this difficulty, a $CO_2$ response signal may still be identifiable, as long as mesocosms that reveal strongest similarities (with respect to initial composition of plankton and their physiological state) are not used as replicates for similar $CO_2$ conditions (perturbations). Instead, mesocosms with similar initial conditions should be exposed to different levels of OA. Such favourable starting conditions were met in the mesocosm experiment described in Engel et al. (2005) and Delille et al. (2005), as well as in the experiment of Eggers et al. (2014).

An alternative approach to setting up mesocosms is to gradually increase the level of perturbation for a series of mesocosms. This way a gradient of different perturbation levels is introduced. The advantage then is that mesocosms that have been collated according to e.g. lowest and highest response signals (or likewise according to similarities in initial conditions) may then be separately analysed with respect to their responses to the individual levels of perturbation.

From this modelling study we infer that collinearities exist between estimates of initial conditions and physiological model parameters, in particular for the photosynthetic efficiencies $\alpha_{phy}$, $\alpha_{cocco}$ and the initial fraction of coccolithophores determined by $f_{cocco}$. Therefore, it is not possible to identify initial concentration of photoautotrophs independently of parameters responsible for phytoplankton growth in HC, MC and LC model solutions. This  was only found because we optimised  initial conditions together with physiological parameters for HC, MC and LC mesocosms separately. By this seperation  we could better specify the $CO_2$ effect on PIC formation.  Doing so we could identify a systematic overestimation of calcification in our model and we conclude that our simulated $CO_2$ effect on PIC formation is even too weak.

**Comment 2:** As a way to reduce the paper in size, I would remove all figures starting from Fig. 11 – and also significantly reduce (or remove) the corresponding text.

**Author's response:** In principle, we see that figures can be remove and we will do so for a revised version of our manuscript. However, to remove all figures starting from Fig. (11) would not be helpful. Some figures are needed to document our findings and to further show the credibility of our model results.

We suggest removing the following figures: 1) Fig. (6, concentrations of dCCHO and TEPC), 2) Fig. (9, ensembles of medium calcification solutions), 3) Fig. (15, fast build-up of photoautotrophic biomass in MC solutions). We find Fig. (14) to be important for the discussion of model bias in PON (exponential growth phase). Figs. (16 and 17) provide relevant information about

the i) underestimated $CO_2$ effect, and the ii) relation between resolved $CO_2$ effect and the full range of variability explained. Figs. (3 and 4) will be merged, which was suggested by the referee, see response to specific comment (8) below.

**Comment 3:** I would also remove the lengthy $CN_{\text{fact}}$ discussion (and the corresponding figures 6 and 7, which are difficult to interpret).

**Author's response:** We will remove Fig. (6) and we will revise text to have the discussion on $CN_{\text{fact}}$ entirely removed, as this would actually go beyond the scope of the study addressed here. Figure (7) will be simplified, by excluding variations of $CN_{\text{fact}}$, and by removing results of the medium calcification (MC) case. We want to retain Fig. (7), because it documents some novel model behaviour with respect to how calcification is simulated in our model approach. We compare this ratio with results from Barcelos e Ramos et al. (2010), which gets somewhat lost in the discussion Sect. (4.1). With the removal of the discussion of Fig. (6) it will become easier to explain and refer to Fig. (7).

*Specific comments by anonymous Referee # 2:*

**Specific comment 1:** Symbol $\theta$ used but not explained.

**Author's response:** The symbol $\theta$ is the chlorophyll-to-carbon ratio (listed in Table 1). We will introduce $\theta$ in the text. The revised text will be moved to the Appendix, where the mathematical notation of the model equations is introduced.

**Specific comment 2:** Equation 13, I would have expected to see zooplankton excretion here.

**Author's response:** We thank the referee for this very thoughtful comment. We realised that we did not state our assumptions in this respect. First of all, we note that it is not a typo in the equations. In our model we are resolving the nitrogen flux of zooplankton excretion but we are eventually not resolving any corresponding net change in TA (total alkalinity). This is because we cannot really differentiate between the excretion of ammonium ($NH_4^+$) and of nitrate ($NO_3^-$) and nitrite ($NO_2^-$). The excretion of one mole $NH_4^+$ would increase TA by one mole, whereas the excretion of one mole $NO_3^-$ or $NO_2^-$ would decrease TA by one mole. In other words, we indirectly impose that excreted N does not include urea and we may assume that half of the N excretion by zooplankton is $NH_4^+$ and the other half is $NO_3^-$ and $NO_2^-$, which would introduce a net TA change of zero. For N assimilation during the exponential growth phase we know that it is primarily driven by the uptake of $NO_3^-$ and $NO_2^-$ (due to initial conditions), and the TA change due to $NH_4^+$ uptake can be neglected. During the post-bloom period we may still have some algal growth, albeit at very low rates, that can be based on the utilisation of $NH_4^+$. In the end, we expect the error because of a neglect of the net TA change induced by zooplankton excretion to be small and limited to the last days of the simulation period.

**Specific comment 3:** Page 13-14, I do not understand how the three simulated mesocosms can be used to estimate data variances ($R_i$) for DIC, as this differs for each mesocosm?

**Author's response:** In our text we did not explain this in detail. However, we realised that it is necessary to do so. Briefly, for the derivation of the standard errors we considered the differences (offsets) between the mean *initial* DIC concentrations of the different $CO_2$ treatments. DIC concentrations of those mesososms that were initially exposed to high $CO_2$ (DIC) concentrations are "offset"-corrected so that their initial mean DIC matches the initial mean of the present day DIC concentrations. Mesocosms of the low $CO_2$ treatment were adjusted likewise. In this manner, all initial mean DIC concentrations have become identical, but changes and variations (between the mesocosms) with respect to these mean values remain. Thus, variances of the respective LC, MC, and HC mesocosms can be calculated after applying these (two) offset corrections to all DIC data of the high- and low $CO_2$ treatements. Eventually, individual standard errors for the LC, MC, and HC mesocosms are derived for all sampling dates. Note that the correlations are computed without any differentiation between LC, MC, and HC mesocosms.

**Specific comment 4:** Page 15, line 2-3: As far as I understand, the proposal distribution is adapted by the AMH algorithm. Thus, the Hessian is used only as the initial proposal.

**Author's response:** Yes, this is correct. To clarify this we will revise the text in our revised version of manuscript. The new text will read:

" The standard deviation information required for generating the initial proposal (Gaussian) distribution in the AMH algorithm is derived from the diagonal elements of Hessian matrix. We approximated the diagonal elements of the Hessian with finite central differences, as described in e.g. Matear (1995), Kidston et al. (2011), and in Kreus and Schartau (2015). To do so we imposed an incremental step size of 1% variation to the respective parameter values. "

**Specific comment 5:** Page 1, line 18: I do not understand what is meant with "optimal" mass flux estimates, as there are two completely different optimisations: one for the data assimilation (DA), one for the optimal resource allocation in phytoplankton.

**Author's response:** By "optimal" mass flux estimates we meant the best solution estimated by our data assimilation approach. To avoid any confusion we suggest changing it to "optimised" mass flux estimates.

**Specific comment 6:** Why is the Chl content not added here?

**Author's response:** There is no particular reason. We will refine Fig. (2) by adding chlorophyll as additional variable to the respective compartments.

**Specific comment 7:** Cumulative plots are difficult to interpret. I would prefer to see the actual probability distributions instead. Also, the figs 3 and 4 could be combined, if the three calcification scenarios for each parameter would be put in the same figure rather than in 3 of them.

**Author's response:** We agree that Figs. (3 and 4) can be combined. However, we do not think that it is better to show the probability densities instead of the cumulative probability distributions. In the end, the cumulative probability distributions are used to derive lower and upper credibility limits of the parameter estimates. Furthermore, if we would prepare the figure with probability densities we would need to resolve differences on the y-axis between the different parameters (i.e. between subplots), unless we normalise all parameter values. This is not needed when plotting the cumulative probability distributions. The parameter values can then be shown as they are, while all y-axis are the same (between zero and one) so that they can be nicely merged.

**Specific comment 8:** Fig 8-10: The main difference is in between low and high calcification scenario – Fig 9. Could be removed.

**Author's response:** Yes, we agree. Main differences are between the HC and LC solutions. Some model solutions of the MC ensemble are close to few solutions of the HC case. Therefore it is not critical to remove current Fig. (9)

All other minor comments and corrections suggested by anonymous Referee # 2 have been implemented.

**Revised manuscript version (with marked changes, blue = text shifted, red = text rewritten or added)**

[revised manuscript text omitted]

with the individual fractions:

$$\text{DetN}_0 = f_{det} \cdot PON_0 \tag{2}$$

$$\text{ZooN}_0 = f_{zoo} \cdot (PON_0 - \text{DetN}_0) \tag{3}$$

$$\text{CoccoN}_0 = f_{cocco} \cdot (PON_0 - \text{DetN}_0 - \text{ZooN}_0) \tag{4}$$

$$\text{PhyN}_0 = (1 - f_{cocco}) \cdot (PON_0 - \text{DetN}_0 - \text{ZooN}_0) \tag{5}$$

For initial zooplankton, coccolithophore and phytoplankton biomass we apply a constant C:N ratio of 6.625. We consider a higher C:N ratio (= $2 \cdot 6.625$) only for initial detritus. Since the mesocosms were filled with post-bloom, nutrient depleted water masses, we assume that all dead particulate organic matter has a C:N ratio that is rather typical for such post-bloom conditions. Initial condition of PIC, DIC, and TA are taken from the data for respective mesocosm, whereas we assume same small fixed values (e.g. DON = 0.05 mmol m$^{-3}$, DOC = 102.5 mmol m$^{-3}$, dCCHO = 1.0 mmol m$^{-3}$ and TEPC = 3.5 mmol m$^{-3}$) as initial conditions for all mesocosms.

**2.3 Design of data assimilation (DA) approach**

A peculiarity of the PeECE-I experiment is that high and low changes in total alkalinity (TA) were found in all three $CO_2$ treatments, in response to differences in calcification (Delille et al., 2005). Because the three distinct patterns in calcification (Fig. 3) are attributable to all three treatments means that a factor other than the $CO_2$ perturbations induced variations between the individual mesocosms. For all other observations no such clear pattern could be identified. We designed our DA approach according to this finding and therefore investigate three possible situations (model solutions) that differ in their TA response: low, medium and high calcification (referred to as LC, MC, and HC respectively). Thus, for each of these three (LC, MC and HC) situations we find  three mesocosms that were subject to three different $CO_2$ levels (initial 700 ppmV, 370 ppmV, and 180 ppmV). By adapting the same nomenclature as in Engel et al. (2005) and in Delille et al. (2005), we can assign the mesocosms M1, M6, and M8 to those with low calcification rates (highest TA), M2, M5, and M7 to the ones with medium calcification and finally M3, M4, and M9 to mesocosms with high calcification rates (lowest TA).

**2.3.1 Definition of cost function (data-model misfit)**

In our DA approach we consider data from the three cases (LC, MC, and HC) separately, but we make identical statistical assumptions. The observation vector ($\boldsymbol{y}_i$) contains daily means of three mesocosms of the following measurements: 1) dissolved inorganic carbon (DIC, mmol m$^{-3}$), 2) dissolved inorganic nitrogen (DIN) (nitrate + nitrite, mmol m$^{-3}$), 3) chlorophyll $a$ (Chl $a$, mg m$^{-3}$), 4) particulate organic nitrogen (PON, mmol m$^{-3}$), 5) particular ogranic carbon (POC, mmol m$^{-3}$), 6) particulate inorganic carbon (PIC, mmol m$^{-3}$), 7) total alkalinity (TA, mmol m$^{-3}$). Like the data vector $\boldsymbol{y}_i$, the vector $H_i(\boldsymbol{x})$ represents mean values of three simulated mesocosms for each calcification case (LC, MC, and HC). It combines results of model states:

C and N biomass concentrations of the photoautotrophs (PhyN & PhyC and CoccoN & CoccoC), of zooplankton (ZooN & ZooC), of detritus (DetN & DetC), and carbon concentration of transparent exopolymers particles (TEPC). The vector of differences ($d_i$) between observation ($y_i$) and model results $H_i(x)$ is given as:

[revised manuscript text omitted]